# ZBP1 not RIPK1 mediates tumor necroptosis in breast cancer

Jin Young Baik[1], Zhaoshan Liu[1], Delong Jiao[1], Hyung-Joon Kwon[1], Jiong Yan[1], Chamila Kadigamuwa[1], Moran Choe[1], Ross Lake[2], Michael Kruhlak[3], Mayank Tandon [4,5], Zhenyu Cai [6], Swati Choksi[1] & Zheng-gang Liu[1 ✉]

Tumor necrosis happens commonly in advanced solid tumors. We reported that necroptosis plays a major role in tumor necrosis. Although several key necroptosis regulators including receptor interacting protein kinase 1 (RIPK1) have been identified, the regulation of tumor necroptosis during tumor development remains elusive. Here, we report that Z-DNA-binding protein 1 (ZBP1), not RIPK1, mediates tumor necroptosis during tumor development in pre-clinical cancer models. We found that ZBP1 expression is dramatically elevated in necrotic tumors. Importantly, ZBP1, not RIPK1, deletion blocks tumor necroptosis during tumor development and inhibits metastasis. We showed that glucose deprivation triggers ZBP1-depedent necroptosis in tumor cells. Glucose deprivation causes mitochondrial DNA (mtDNA) release to the cytoplasm and the binding of mtDNA to ZBP1 to activate MLKL in a BCL-2 family protein, NOXA-dependent manner. Therefore, our study reveals ZBP1 as the key regulator of tumor necroptosis and provides a potential drug target for controlling tumor metastasis.

[1] National Cancer Institute; National Institutes of Health, Laboratory of Immune Cell Biology, Bethesda, MD, USA. [2] National Cancer Institute; National Institutes of Health, Laboratory of Genitourinary Cancer Pathogenesis, Bethesda, MD, USA. [3] National Cancer Institute; National Institutes of Health, Laboratory of Cancer Biology and Genetics, Bethesda, MD, USA. [4] National Cancer Institute; National Institutes of Health, Collaborative Bioinformatics Resource, Bethesda, MD, USA. [5] Advanced Biomedical Computational Science, Frederick National Laboratory for Cancer Research, Frederick, MD, USA. [6] Tongji University Cancer Center, Shanghai Tenth People's Hospital, School of Medicine, Tongji University, Shanghai, China. ✉email: zgliu@helix.nih.gov

Necrosis is a form of cell death distinct from apoptosis. While apoptotic cells normally have plasma membrane blebbing and nuclear fragmentation and retain membrane integrity, necrotic cells bear the character of rounding of the cell, gaining in cell volume, the rupture of plasma membrane and the release of intracellular contents[1–4]. Necroptosis is a programmed necrosis[1–5]. By studying TNF-induced necroptosis, our understanding of the molecular mechanism of necroptosis has been greatly improved. It is now known that receptor interacting protein kinase 1 and 3 (RIPK1 and RIPK3) and the mixed lineage kinase domain-like (MLKL) constitute the core of the necroptosis machinery[2–5]. Importantly, RIPK3-mediated phosphorylation of MLKL results in oligomerization of the latter and its subsequent translocation to the plasma membrane[6–9]. While this pathway is also critical for other death factors, such as FasL or TRAIL, -induced necroptosis, RIPK1 is not involved in viral infection and Toll-like receptor (TLR)-mediated necroptosis, although RIPK3 and MLKL seem to be the common players of necroptosis[5]. Two other proteins, Z-DNA-binding protein 1 (ZBP1), also known as DAI/DLM-1, and TIR-domain-containing adapter-inducing interferon-β (TRIF) are known to function upstream of RIPK3 and interact with RIPK3 through their RHIM (RIP homotypic interaction motif) domains to mediate necroptosis in response to viral infection or TLR signaling respectively[10,11].

Although recent studies have reported that the necroptotic pathway may be involved in tumorigenesis[12], it is not clear if tumor cells undergo necroptosis during tumor development, and if so, what role necroptosis of tumor cells plays in tumorigenesis and tumor metastasis. Foci of cell death are commonly observed in core regions of solid tumors as a result of inadequate vascularization and subsequent metabolic stresses such as hypoxia and nutrient deprivation[13,14]. Because the morphology of dead tumor cells appears to be necrotic, these foci of cell death are referred as tumor necrosis[12,15,16]. Recently, we demonstrated that tumor necrosis is mainly necroptotic[17]. Tumor necrosis is thought to be caused by hypoxia and nutrient deprivation[13,14], but it is not clear whether these conditions lead to necroptosis of tumor cells. On the other hand, some recent studies suggested that tumor necroptosis may be triggered by death factors and engages RIPK1 and RIPK3 pathway during tumor development[18,19]. However, a recent study showed that RIPK1 is needed for inflammatory diseases, but not tumor growth or metastasis[20]. Therefore, the underlying mechanism of tumor necrosis during tumorigenesis remains elusive. Particularly, the role of RIPK1 in tumor necrosis has not been directly evaluated. In this current study, we report that Z-DNA-binding protein 1 (ZBP1), not RIPK1, mediates necroptosis of tumor cells during tumor development in preclinical cancer models. We found that the levels of ZBP1 and RIPK3 expression are dramatically elevated in necrotic tumors during tumor development. Importantly, we demonstrated that deletion of ZBP1 blocks necroptosis of tumor cells during tumor development and consequently inhibits tumor metastasis in MVT-1 breast cancer model. In addition, we showed that glucose deprivation (GD), not hypoxia or glutamine deprivation (GlnD), triggers ZBP1-depedent necroptosis. Therefore, our study reveals ZBP1 as the key regulator of tumor necroptosis during tumor development.

## Results

**RIPK1 is not required for MVT-1 tumor necroptosis.** To investigate the role of RIPK1 in tumor necroptosis, we first tested if the specific RIPK1 inhibitor, Necrostatin-1 (Nec-1), has any effect on tumor necroptosis in the MVT-1 mouse breast cancer model[21,22] as others and our early studies showed that Nec-1 effectively blocks TNF-induced necroptosis in vivo[1,23,24]. In our

previous study, we demonstrated that the status of tumor necrosis and MLKL phosphorylation, which happens mainly in tumor cells, reflects the level of tumor necroptosis in solid tumors[17]. We found that tumor necrosis and MLKL phosphorylation were not affected by the administration of Nec-1 (Fig. 1a, b). These results imply that necroptosis of tumor cells may be independent of RIPK1. To confirm this possibility, we generated RIPK1 knockout (RIPK1 KO) and CRISPR control (CRISPR CT) MVT-1 cells with CRISPR/Cas9 technology (Supplementary Fig. 1a). RIPK1 deletion does not affect the in vitro and in vivo proliferation of MVT-1 cells and MVT-1 tumor growth (Supplementary Fig. 1b–d). Importantly, we found that RIPK1 deletion actually increased, instead of reducing, tumor necrosis and MLKL phosphorylation in RIPK1 KO tumors (Fig. 1c, d). The deletion of RIPK1 protein does not affect the expression levels of MLKL and RIPK3 in tumors (Fig. 1e). Consistent with our previous report[17], necroptosis is the predominant form of cell death in tumor necrosis areas as there are limited apoptotic cells in both CRISPR CT and RIPK1 KO tumors (Supplementary Fig. 1e). Therefore, these data suggest that RIPK1 is not required to undergo necroptosis during tumor development in MVT-1 breast cancer model.

**ZBP1 is highly expressed in necrotic tumors.** Since RIPK1 is not essential for necroptosis of tumor cells during tumor development, we searched for possible tumor necroptosis mediators upstream of RIPK3. We examined the expression levels of two other known RIPK3 upstream effectors of necroptosis, Z-DNA-binding protein 1 (ZBP1), also known as DAI/DLM-1, and TIR-domain-containing adapter-inducing interferon-β (TRIF) in MVT-1 and MMTV-PyMT tumors, a genetically engineered mouse model (GEMM) of breast cancer[10,11,25]. We found that the expression levels of ZBP1 are significantly increased in the later stages of these tumors when necrosis happens (Fig. 2a, b)[17]. The expression levels of RIPK3 protein are also elevated. However, neither RIPK1 nor TRIF expression level is increased in these tumors (Fig. 2a, b and Supplementary Fig. 2a, b). To exclude the possibility that this increase of ZBP1 expression is due to infiltrating immune cells that are known to have ZBP1 expression[26], we generated GFP-expressing MVT-1 cells and isolated pure population of tumor cells (Supplementary Fig. 2c, d). Consistent with our findings with tumor lysates, the levels of ZBP1 and RIPK3, but not RIPK1, are dramatically increased in GFP-MVT-1 cells isolated from tumors compared to that in cultured cells (Fig. 2c). Interestingly, RIPK1 protein level is decreased in the isolated tumor cells. The increased expression of ZBP1 and RIPK3 also happens in mouse melanoma B16, mouse lung cancer LLC, human breast cancer MCF-7 and MDA-MB-231 tumors as their expression levels are significantly higher in the cells from tumors compared to that in the cultured parental cells (Fig. 2d, e and Supplementary Fig. 2e, f). These results confirmed that ZBP1 and RIPK3 proteins are also highly expressed in other types of solid tumors. In addition, we found that RIPK1 protein levels are significantly decreased in these tumors as well. While there is a modest decrease of RIPK1 mRNA level in tumors, the mRNA levels of ZBP1 and RIPK3 are significantly increased in isolated tumor cells (Fig. 2f). Therefore, these results suggest that the expression of ZBP1 and RIPK3 are increased at the advance stages of solid tumors. Through bioinformatic analysis of the human cancer datasets, we found that human ZBP1 expression is significantly increased in human breast cancer and several other types of solid tumors when compared to their normal tissue (Fig. 2g and Supplementary Fig. 2g, h). Previous studies have suggested that RIPK3 expression is low in primary tumors due to DNA methylation[27]. In both MVT-1 and MMTV-PyMT tumors (Fig. 2a, b), while RIPK3 levels are very low/undetectable in the

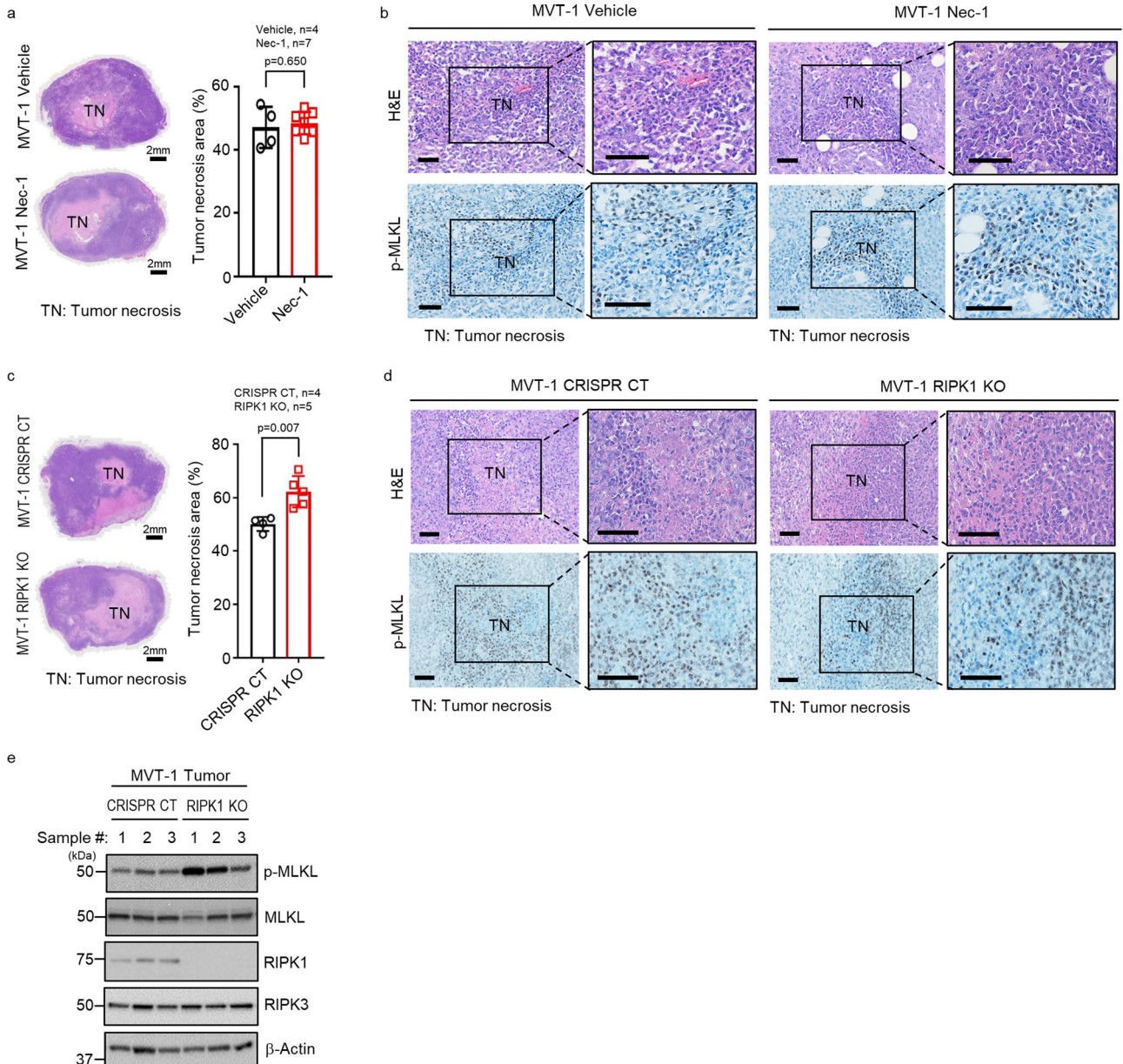

**Fig. 1 RIPK1 is not required for MVT-1 mammary tumor necroptosis. a** FVB/NJ mice at three weeks post-implantation with MVT-1 cells were treated weekly with vehicle or Necrostatin-1 (Nec-1; *i.v.*) until week 5. Left panel shows the representative images of H&E stained 5-week tumors of vehicle or Nec-1 treated mice. Scale bar, 2 mm. Right panel shows the percentage of tumor necrosis area (TN) of the total tumor area from mice at 5-week. Data are presented as mean values ± SEM. **b** Representative H&E and immunohistological images of phospho-MLKL (p-MLKL) antibody staining of tumor sections from mice implanted and treated as in Fig. **a**. Scale bar, 50 μm. **c** MVT-1 CRISPR/Cas9 control (CRISPR CT) or MVT-1 RIPK1 knock out (RIPK1 KO) cells were generated by using the CRISPR/Cas9 system. Left panel shows the representative H&E image of 4-week tumors from FVB/NJ mice implanted with the MVT-1 CRISPR CT or MVT-1 RIPK1 KO cells. Scale bar, 2 mm. Right panel shows the percentage of tumor necrosis area (TN) of the total tumor area after 4-weeks. Data are presented as mean values ± SEM. **d** Representative H&E and immunohistological images of p-MLKL antibody staining of 4-week tumor sections from mice implanted as in Fig. **c**. Scale bar, 50 μm. **e** Western blotting analysis of tumor lysates from mice implanted as in Fig. **c** using the indicated antibodies. Western blotting analysis representative of three independent experiments. Two-sided student's *t* test was used to determine the statistical significance of differences between groups. Differences with *P* values < 0.05 were considered significant. Source data are provided as a Source Data file.

early stages of tumor development, RIPK3 expression is dramatically upregulated when tumor reaches certain sizes. We found that DNA methylation of RIPK3 and DNA methyltransferase 1 (DNMT1) are dramatically decreased in late stage MVT-1 tumors, suggesting that the reprogramming of RIPK3 expression is likely due to the reduction of DNA methylation (Fig. 2h, Supplementary Fig. 2i).

**ZBP1 is essential for tumor necroptosis**. To examine whether ZBP1 is needed for tumor necroptosis during MVT-1 mammary tumor development, we generated CRISPR ZBP1 knockout (ZBP1 KO) and CRISPR control (CRISPR CT) MVT-1 cells by transient transfection and selected clones with no Cas9 expression for our study because Cas9 is immunogenic[28]. We found that ZBP1 deletion does not affect in vitro and in vivo proliferation of

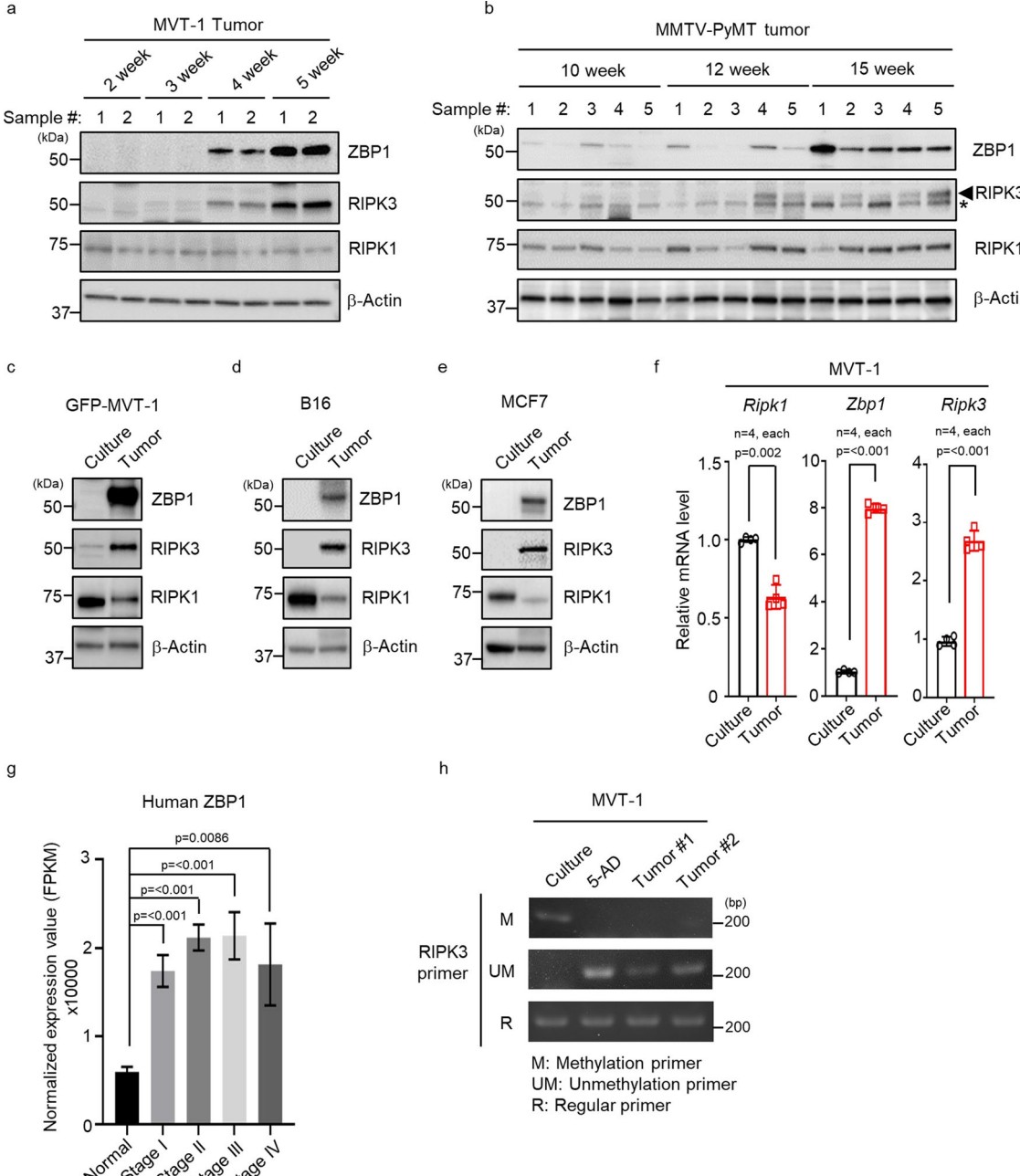

**Fig. 2 ZBP1 is highly increased in late stage of mouse and human tumors. a–e** Western blotting analysis representative of three independent experiments. **a** MVT-1 tumors were collected at 2–5 weeks post-implantation and tumor cell lysates were analyzed by western blotting with the indicated antibodies. **b** MMTV-PyMT breast tumors were collected at 10–15 weeks and tumor cell lysates were analyzed by western blotting with the indicated antibodies (*, non-specific band). **c** GFP-MVT-1 tumor cells were collected at 5-week post implantation and the lysates were analyzed by western blotting with the indicated antibodies. **d** C57BL/6 J mice implanted with syngeneic B16 mouse melanoma cells and tumor cells were collected at 2-weeks post-implantation. Western blot analysis of cultured B16 cell lysates or B16 tumor cell lysates was done using the indicated antibodies. **e** BALB/c-nu/nu mice implanted with MCF7 human breast cancer cells and tumor cells were collected at 8-weeks post-implantation. Western blot analysis of cultured MCF7 cell lysates or MCF7 tumor cell lysates was done using the indicated antibodies. To be comparable to the late stage necrotic tumors, the tumors from these models in **d** and **e** were collected when they approached 1500–2000 mm³ in volume and had tumor necrosis. **f** Quantitative real-time PCR analysis of the relative expression of *Ripk1* or *Zbp1* or *Ripk3* mRNA from cultured MVT-1 cells or MVT-1 tumor cells isolated from the mice implanted with MVT-1 cells and collected at 5 weeks ($n = 4$, each). Data are presented as mean values ± SEM. **g** Differential expression of human ZBP1 was analyzed for human breast cancer dataset (TCGA-BRCA) and performed for each tumor stage (Normal stage, $n = 112$; Stage I, $n = 182$; Stage II, $n = 627$; Stage III, $n = 249$; Stage IV, $n = 20$). Data are presented as mean values ± SD. **h** Methylation-specific PCR of genomic DNA from MVT-1 or 5-Aza-2′-deoxycytidine (5-AD), treated MVT-1 or 5-week MVT-1 tumor was detected by using methylation primer (M) or unmethylation primer (UM) or regular primer (R) for RIPK3. The PCR result is representative of three independent experiments. Two-sided student's *t* test was used to determine the statistical significance of differences between groups. Differences with *P* values < 0.05 were considered significant. Source data are provided as a Source Data file.

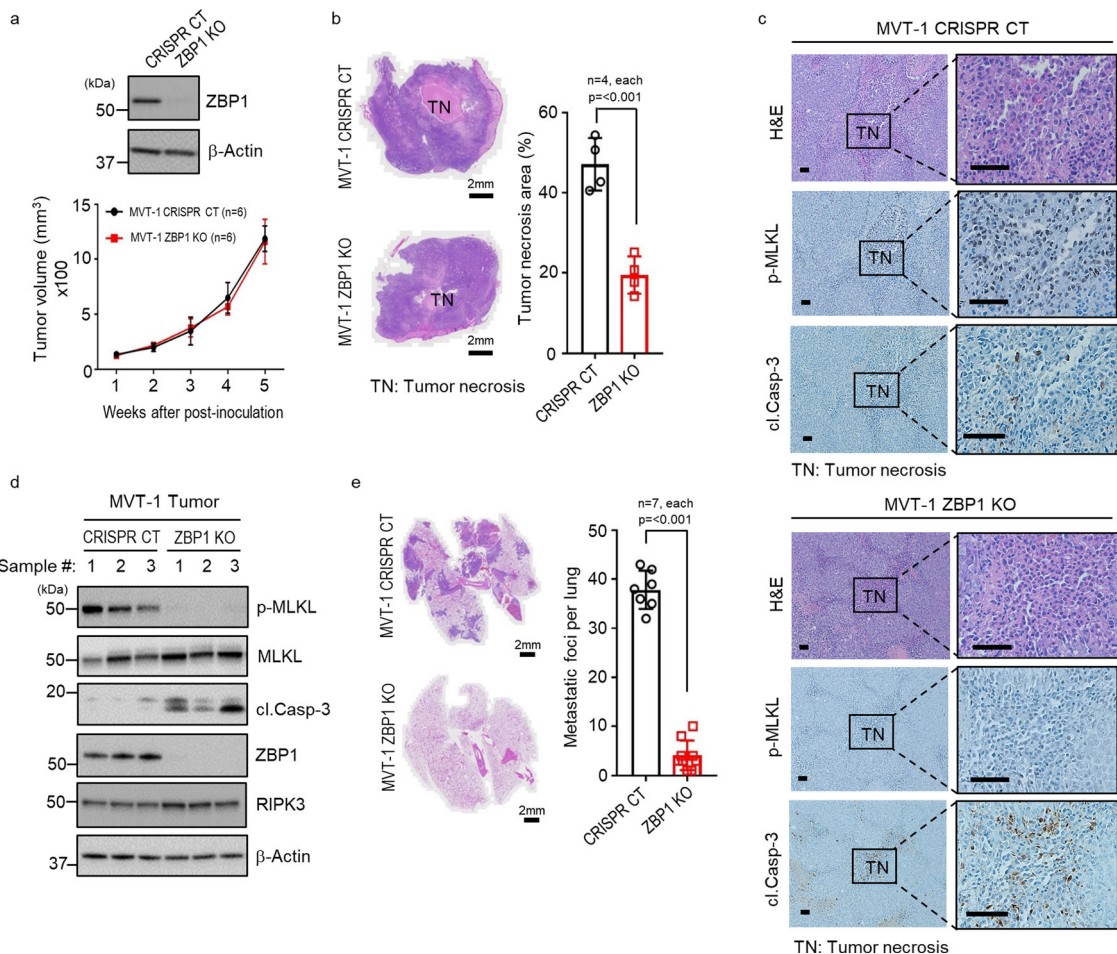

**Fig. 3 ZBP1 is critical for tumor necroptosis and lung metastasis. a** Western blotting analysis representative of three independent experiments. MVT-1 CRISPR/Cas9 control (CRISPR CT) or ZBP1 knock out (ZBP1 KO) tumor lysates were blotted using the indicated antibodies (upper panel). Tumor growth curve by measuring tumor volume of FVB/NJ mice implanted with MVT-1 CRISPR CT or ZBP1 KO cells (lower panel). **b** Left panel shows the representative images of H&E stained tumors at 5-weeks post-implantation as in Fig. **a**. Scale bar, 2 mm. Right panel shows the percentage of tumor necrosis area (TN) of the total tumor area from mice at 5-weeks post-implantation. Data are presented as mean values ± SEM. **c** Representative images of H&E and immunohistological stained sections with phospho-MLKL (p-MLKL) or cleaved caspase-3 (cl.Casp-3) antibodies of 5-week tumor sections from mice implanted as in Fig. **a**. Scale bar, 50 µm. **d** Western blotting analysis is representative of three independent experiments. 5-week tumor lysates from mice implanted as in Fig. **a** was determined by using the indicated antibodies. **e** Left panel shows the representative images of H&E stained lung sections from mice implanted as in Fig. **a** showing lung metastasis. Scale bar, 2 mm. Right panel shows the quantification of metastatic foci in lungs from mice at 5-week post-implantation. Data are presented as mean values ± SEM. Two-sided student's *t* test was used to determine the statistical significance of differences between groups. Differences with *P* values < 0.05 were considered significant. Source data are provided as a Source Data file.

MVT-1 cells and MVT-1 tumor growth (Fig. 3a and Supplementary Fig. 3a, b). When examining the levels of necroptosis of the tumors from these cells, we found that tumor necrosis is dramatically reduced in ZBP1 KO tumors compared to CRISPR CT tumors by comparing the middle sections of four different WT and ZBP1 KO tumors and the serial sections of three different pairs of WT and ZBP1 KO tumors (Fig. 3b and Supplementary Fig. 3d) and that, while there is abundant MLKL phosphorylation in necrotic areas of CRISPR CT tumors, little MLKL phosphorylation is detected in ZBP1 KO tumors (Fig. 3c). In contrast, the cleaved Casp-3 positive and TUNEL positive cells in ZBP1 KO tumors are dramatically increased compared to CRISPR CT tumors, indicating blocking necroptosis leads to the increase of apoptosis as we reported previously[17] (Fig. 3c and Supplementary Fig. 3c). These observations with IHC are confirmed in the ZBP1 KO tumor lysates by western blot analysis (Fig. 3d). These results suggest that ZBP1 plays a critical role in tumor necroptosis in MVT-1 breast cancer model. These findings were confirmed using a different ZBP1 sgRNA to knock out ZBP1

in MVT-1 cells (Supplementary Fig. 3e, f). This conclusion is further supported by knocking down of ZBP1 in B16 mouse melanoma tumor model (Supplementary Fig. 3g, h). We found that ZBP1 knockdown leads to the reduction of tumor necrosis and MLKL phosphorylation, but increases of cleaved Casp-3 and TUNEL positive cells in B16 melanoma tumors (Supplementary Fig. 3i, k). Therefore, ZBP1 is essential for tumor necroptosis during tumor development in these cancer models.

The MVT-1 breast cancer model is known to develop lung metastasis at 4 to 5 weeks post-implantation[22]. Examining the effect of ZBP1 deletion on lung metastasis, we found that mice implanted with CRISPR CT tumor cells had high lung metastasis whereas mice with ZBP1 KO MVT-1 cells had little lung metastasis (Fig. 3e). These data were confirmed using the second ZBP1 sgRNA KO cells (Supplementary Fig. 3l). ZBP1 deletion does not affect MVT-1 cells' ability to migrate and extravasation (Supplementary Fig. 3m, n). Therefore, these data suggest that ZBP1 deletion dramatically reduces lung metastasis in MVT-1 breast cancer model.

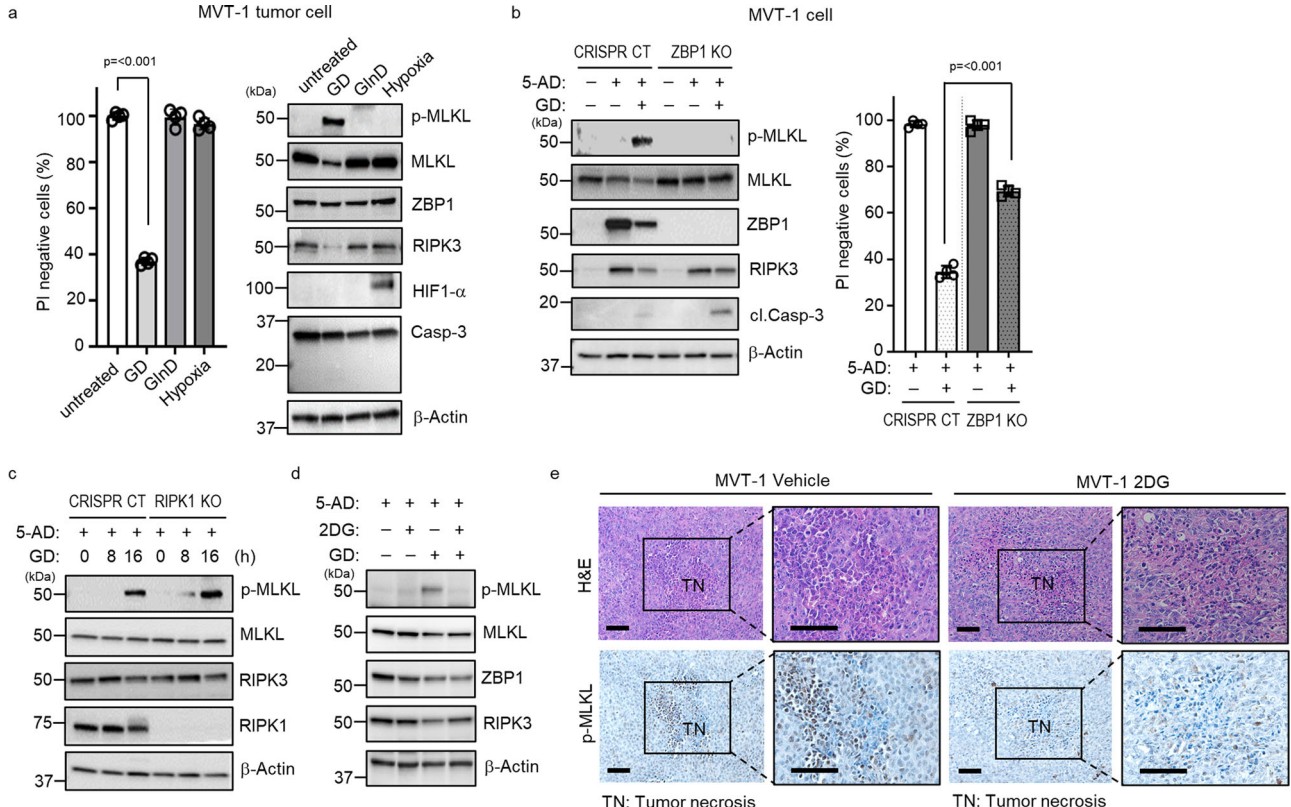

**Fig. 4 Glucose deprivation (GD) induces tumor necroptosis in vitro and in vivo. a** Primary MVT-1 tumor cells were isolated at 5 weeks post-implantation and treated for 36 h with 0.5 mM glucose (GD) or 0.1 mM glutamine (GlnD) or 0.1% O₂ (hypoxia) and cell death was determined by PI staining using flow cytometry (left panel, n = 4 biologically independent samples, each) or representative of three independent experiments analyzed by western blotting using the indicated antibodies (right panel). Data are presented as mean values ± SEM. **b** Representative of three independent western blotting analysis. MVT-1 CRISPR CT or MVT-1 ZBP1 KO cells were treated with 5-Aza-2′-deoxycytidine (5-AD), for 3 days, followed by GD condition for 24 h (left panel) and determined by using the indicated antibodies. Cell death analysis of the cells treated with 5-AD for 3 days, followed by GD condition for 30 h, was determined by PI staining using flow cytometry (right panel, n = 4 biologically independent samples, each). Data are presented as mean values ± SEM. **c** Representative of three independent western blotting analysis. MVT-1 CRISPR CT or MVT-1 RIPK1 KO cells were treated with 5-AD for 3 days, followed by GD condition for the indicated time points and determined by using the indicated antibodies. **d** Representative of three independent western blotting analysis. MVT-1 cells were treated with 5-AD for 3 days, followed by treatment with 2-Deoxy-D-glucose (2DG) and determined by using the indicated antibodies. **e** FVB/NJ mice at three weeks post-implantation with MVT-1 cells were treated with vehicle or 2DG (i.p.) daily until week 5. Representative images of H&E and immunohistological stained sections with phospho-MLKL (p-MLKL) antibody. Scale bar, 50 μm. Two-sided student's t test was used to determine the statistical significance of differences between groups. Differences with P values < 0.05 were considered significant. Source data are provided as a Source Data file.

**Glucose deprivation induces ZBP1-dependent necroptosis.** RIPK1 is essential for death factor-induced necroptosis[2–5,29,30]. As we demonstrated that RIPK1 is dispensable for tumor necroptosis, it is unlikely that death factors including TNF are the key triggers in tumor necroptosis. On the other hand, as necroptosis is crucial for tumor necrosis and tumor necrosis is thought to be caused by hypoxia and nutrient deprivation[13,14], we then tested whether hypoxia, glucose deprivation (GD) or glutamine deficiency (GlnD) could induce necroptosis in tumor cells. We found that GD, but not GlnD or hypoxia, induced MVT-1 and MMTV-PyMT tumor cells to undergo necroptosis, without caspase-3 activation within 36 h (Fig. 4a and Supplementary Fig. 4a–c). Extended treatment of GlnD or hypoxia for 72 h and longer was needed to trigger apoptosis, not necroptosis, in these tumor cells (Supplementary Fig. 4d, e). In addition, as interferons (IFNs) are known to induce tumor cell death and ZBP1 expression[31], we also checked if IFNs could induce necroptosis in these tumor cells and found that IFNs do not induce MLKL phosphorylation, but some Casp-3 cleavage (Supplementary Fig. 4f, g). Since necroptosis is the dominant form of tumor cell death in necrotic tumors, these results indicate that GD is the

possible trigger for tumor necroptosis during tumor development. We next tested whether ZBP1 is required for GD-induced necroptosis in MVT-1 cells. Because the levels of ZBP1 and RIPK3 proteins are quite low in cultured MVT-1 cells (Supplementary Fig. 4h), we need to elevate the expression levels of these proteins before GD treatment. To restore the RIPK3 expression, we treated cells with the DNA methyl-transferase inhibitor, 5-AD (5-Aza-2′-deoxycytidine), and found that not only RIPK3 expression is increased, but surprisingly, the ZBP1 protein level is also dramatically elevated (Fig. 4b and Supplementary Fig. 4h). Importantly, we found that GD treatment induced MLKL phosphorylation in CRISPR CT, but not in ZBP1 KO cells and at the same time, while GD treatment leads to little Casp-3 activation in CRISPR CT cells, Casp-3 cleavage is increased in ZBP1 KO cells (Fig. 4b). Consistently, ZBP1 deletion significantly reduced GD-induced cell death (Fig. 4b). We verified the role of ZBP1 in GD-induced necroptosis by knocking down ZBP1 with shRNA in MVT-1 cells (Supplementary Fig. 4i) as well as in B16 cells (Supplementary Fig. 4j). Importantly, we confirmed this observation with CRISPR CT and ZBP1 KO cells from MVT-1 tumors without 5-AD treatment (Supplementary Fig. 4k). Additionally,

we found that IFN-induced apoptosis is RIPK1-independent in MVT-1 cells (Supplementary Fig. 4l). These results suggest that ZBP1 is essential for GD-induced necroptosis in tumor cells. Consistent with our early finding that RIPK1 is not required for tumor necroptosis during tumor development, we found that RIPK1 is not required for GD-induced MLKL phosphorylation (Fig. 4c). As shown in Supplementary Fig. 4a, GD triggers a graduate decrease of RIPK3 protein while MLKL phosphorylation is increasing. We confirmed that the decrease of RIPK3 protein is due to protein degradation as the presence of the proteasome inhibitor MG132 blocks the GD-induced decrease of RIPK3 (Supplementary Fig. 4m).

The glucose analog, 2-Deoxy-D-glucose (2DG), is known to block glycolysis at the phosphoglucoisomerase level because it cannot undergo further glycolysis[32]. Since it has been shown that 2DG blocks GD-induced cell death[33], we therefore tested if 2DG inhibits GD-induced necroptosis in MVT-1 cells. We found that 2DG treatment blocks GD-induced MLKL phosphorylation in MVT-1 cells (Fig. 4d). More importantly, as a previous study reported that 2DG has limited effect on tumor growth in the orthotopic breast tumor model[32], we then examined if 2DG blocks tumor necroptosis in our MVT-1 tumor model by administering 2DG, daily for two weeks, to mice 3 weeks after implantation of MVT-1 cells. As shown in Fig. 4e, we found that 2DG treatment significantly reduced MLKL phosphorylation in MVT-1 tumors. This result indicates that glucose deprivation plays a key role in triggering tumor necroptosis during tumor development.

**The Z-DNA binding domain 2 (Zα2) of ZBP1 is critical for ZBP1 to mediate tumor necroptosis.** ZBP1 is known to interact with RIPK3 to mediate viral infection-induced necroptosis[11]. We examined if GD treatment induces ZBP1 and RIPK3 interaction. Due to the lack of suitable anti-ZBP1 antibody for immunoprecipitating endogenous ZBP1 protein, we generated HA-ZBP1 MVT-1 cells by stably expressing HA-ZBP1 protein and performed immunoprecipation experiment with anti-HA antibody. After 5-AD treatment that elevates RIPK3 expression in these cells, HA-ZBP1 protein was immunoprecipitated from cells collected at different time points after GD treatment. As shown in Fig. 5a, RIPK3 protein was co-precipitated with HA-ZBP1 after 12 h of GD treatment, suggesting that GD treatment triggers the formation of ZBP1/RIPK3 complex. For viral infection-induced necroptosis, it has been reported that the Z-DNA binding domain 2 (Zα2) of ZBP1 is critical for sensing the viral nucleic acids[34]. To gain some insight into how ZBP1 senses the signal from GD treatment, we generated Zα2 ZBP1 mutant protein and tested if the Zα2 domain is critical for ZBP1 to mediate GD-induced MLKL phosphorylation (Fig. 5b). Since the RHIM A domain of ZBP1 is known to mediates the interaction with RIPK3, we also mutated this domain as a control for the loss of function of ZBP1 protein. We then transfected the WT and these mutant ZBP1 plasmids into ZBP1 KO MVT-1 cells and examined if they could restore GD-induced MLKL phosphorylation. As shown in Fig. 5b, while WT ZBP1 is able to restore GD-induced MLKL phosphorylation in ZBP1 KO MVT-1 cells, both Zα2 and RHIM A ZBP1 mutants failed to do so. These results suggested that Zα2 domain is critical for ZBP1 to mediate GD-induced necroptosis in MVT-1 cells.

To evaluate if the Zα2 domain is critical for ZBP1 to mediate tumor necroptosis during tumor development, we implanted MVT-1 CT, ZBP1 KO, and WT or different mutant ZBP1 reconstituted KO cells into the fat pads of mice and examined the status of MLKL phosphorylation and Casp-3 cleavage in the resulting tumors of these different cells. As shown in Fig. 5c, while WT ZBP1 protein restored the MLKL phosphorylation that is absent in ZBP1 KO tumors, the Zα2 mutant ZBP1 protein failed to do so. The presence of the WT, not the Zα2 mutant, ZBP1 protein also reduced the cleavage of Casp-3 in tumor cells. These observations were confirmed by western blotting of different tumor samples (Fig. 5d). We examined the lung metastasis in mice implanted with WT or Zα2 mutant ZBP1 reconstituted ZBP1 KO MVT-1 cells. While WT ZBP1 restored the metastasis of ZBP1 KO cells, Zα2 mutant ZBP1 failed to do so (Fig. 5e). This data suggests that the Z-DNA binding domain of ZBP1 is required for its function in promoting metastasis. Therefore, these results suggested that the Zα2 domain of ZBP1 is critical for ZBP1 to mediate tumor necroptosis during tumor development and imply that ZBP1 may need to sense some nucleic acids signal triggered by GD to engage necroptosis.

**Release of mitochondrial DNA needed for GD-induced necroptosis.** It is reported that mitochondrial DNA (mtDNA) is released to the cytosol and binds to ZBP1 under stress conditions[35]. To test if cytosolic mtDNA is involved in GD-induced necroptosis, we first examined if mtDNA is released under GD condition. The enhanced cytosolic DNA staining and the reduction of mtDNA mitochondrial localization indicated that mtDNA may be released into the cytosol in MVT-1 and tumor cells from MVT-1 or MMTV-PyMT under GD conditions (Fig. 6a and Supplementary Fig. 5a–c). We confirmed that mtDNA is indeed released into the cytoplasm under GD condition by detecting the presence of mitochondrial genes, cytochrome B (*CytB*) and NADH dehydrogenase 2 (*Nd2*), in the cytosolic DNAs (Fig. 6b). We examined whether mtDNA is released in MVT-1 tumor paraffin sections of 2-week or 4-week from mice implanted with MVT-1 cells by co-staining mitochondria (green)[36] and mitochondrial DNA (red)[37]. The data suggest that mtDNA may be relased from mitochondria in some tumor cells of 4-week tumor (Supplementary Fig. 5d), implying that mtDNA may be released to the cytosol in tumors that bear necroptosis. We found that the cytosolic mtDNA binds to ZBP1 as the mitochondrial *CytB* and *Nd2* DNAs were co-precipitated with ZBP1 under GD condition with or without 5-AD (Fig. 6c and Supplementary Fig. 5e). The 5-AD treatment alone does not induce the release of mtDNA to the cytosol (input) and mtDNA binding to ZBP1 (IP) while GD alone is sufficient to trigger the release and binding. However, the presence of 5-AD made the cells more sensitive to GD and the release of mtDNA as shown (input), likely the toxicity of 5-AD made cells more fragile. Simillar results were obtained with the immunoprecipitation of endogenous ZBP1 (Supplementary Fig. 5f). Importantly, transfection of the cytosolic DNAs isolated from GD-treated MVT-1 cells induces ZBP1-mediated MLKL phosphorylation (Fig. 6d). Because a previous report suggested that the cytosolic DNA sensing pathway, cGAS/STING, may be involved in necroptosis[38], we knocked out STING in MVT-1 cells and found that STING deletion has no effect on GD-induced MLKL phosphorylation (Supplementary Fig. 5g). To further demonstrate the role of the cytosolic mtDNA in GD-induced necroptosis, we explore whether the release of mtDNA is critical for this process. As mitochondrial proteins, NOXA and PUMA, are implied in regulating mitochondrial integrity under different cell stress conditions[35,39], we then tested if these proteins are involved in the engagement of necroptosis in response to GD. We found that that PUMA knockdown has no effect on GD-induced MLKL phosphorylation (Supplementary Fig. 5h). However, NOXA knockdown by either siRNA or shRNA dramatically reduced GD-induced cytosolic release of mtDNAs and

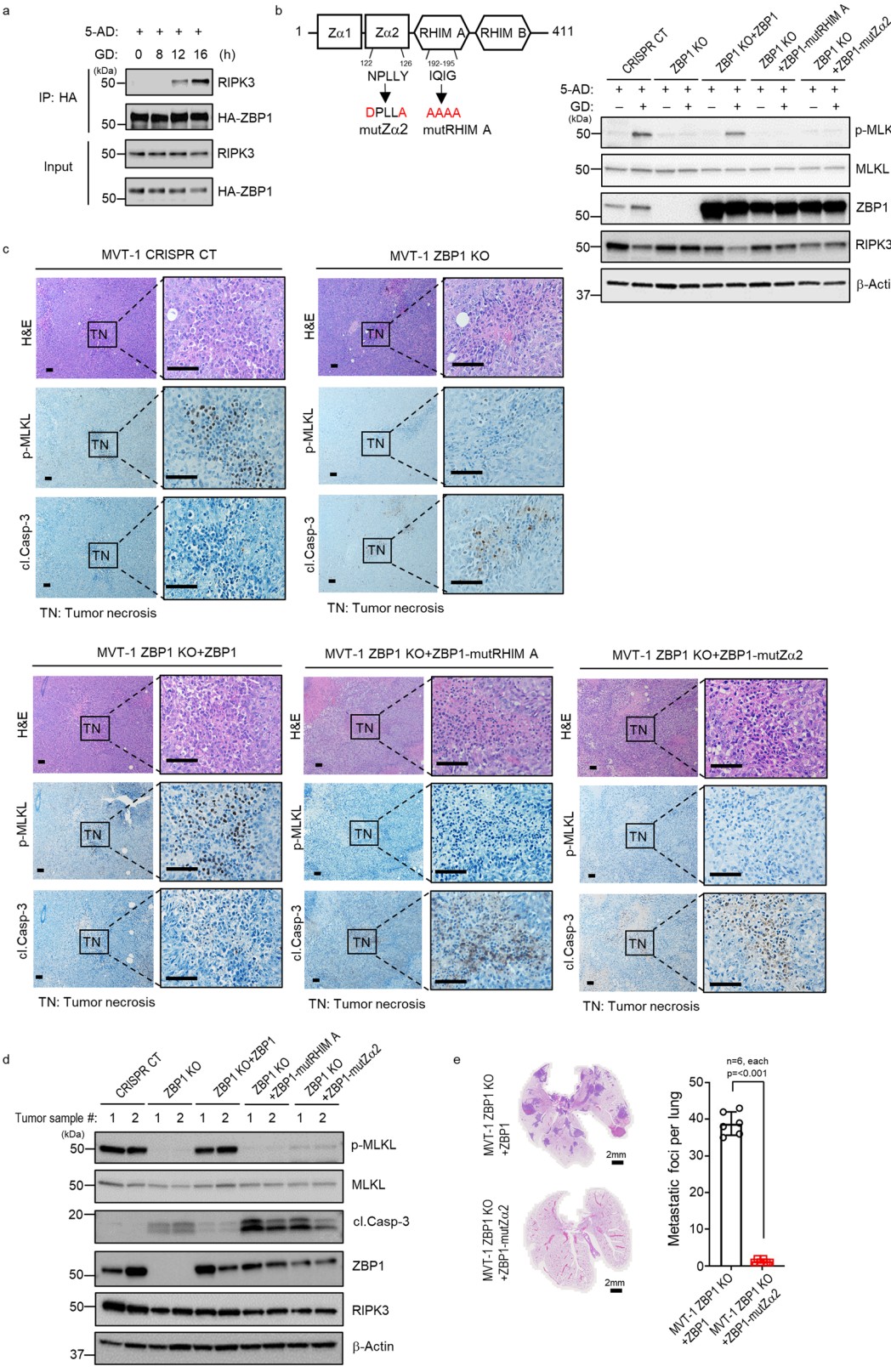

MLKL phosphorylation in MVT-1 cells (Fig. 6e–g and Supplementary Fig. 5i). The effect of NOXA knockdown on GD-induced cytosolic release of mtDNAs and MLKL phosphorylation was also confirmed in B16, Met-1 and MCF7 cells (Supplementary Fig. 5j–n). Taken together, these data indicate that the mtDNA is released from mitochondria under GD and the cytosolic mtDNAs are critical for triggering GD-induced necroptosis.

## Discussion

Evasion of apoptosis is a hallmark of cancer and is critical for tumor development[40]. As programmed necrosis, the role of necroptosis in tumorigenesis has been investigated and both tumor-promoting and tumor-suppressing effects of necroptosis on tumor development were reported[41,42]. Therefore, understanding the mechanism of tumor necroptosis will be critical for

**Fig. 5 The Z-DNA binding domain 2 (Zα2) of ZBP1 is critical for mediating tumor necroptosis. a** Immunoprecipitation and western blotting analysis representative of three independent experiments. MVT-1 cells stably overexpressing HA-ZBP1 were treated with 5-Aza-2′-deoxycytidine (5-AD), for 3 days followed by glucose deprivation (GD) condition for the indicated time points. Cell lysates were immunoprecipitated with HA antibody and the immunoprecipitated complexes were immunoblotted. **b** Structure of mouse ZBP1 protein indicating the strategy for RHIM A mutant or Zα2 mutant (left panel). Western blot analysis representative of three independent experiments of MVT-1 ZBP KO cells transfected with reconstituted WT ZBP1 or RHIM A mutant or Zα2 mutant. The transfected cells were then treated with 5-AD for 3 days, followed by GD condition for 24 h, by using the indicated antibodies (right panel). **c** FVB/NJ mice were implanted with the MVT-1 CRISPR CT or MVT-1 ZBP1 KO or ZBP1 KO reconstituted with WT ZBP1 or RHIM A mutant or Zα2 mutant protein. Representative images of H&E and immunohistological stained sections with phospho-MLKL (p-MLKL) or cleaved caspase 3 (cl.Casp-3) antibody of each tumors. Scale bar, 50 μm. **d** Western blotting analysis representative of three independent experiments of 5-week tumor lysates from mice implanted as in Fig. **c**. The tumor lysates were determined by using the indicated antibodies. **e** FVB/NJ mice were implanted with the MVT-1 ZBP1 KO cells reconstituted with WT ZBP1 or Zα2 mutant protein. Left panel shows the representative images of H&E stained lung sections from mice showing lung metastasis. Scale bar, 2 mm. Right panel shows the quantification of the metastatic foci in lungs from mice at 5-week post-implantation. Data are presented as mean values ± SEM. Two-sided student's *t* test was used to determine the statistical significance of differences between groups. Differences with *P* values < 0.05 were considered significant. Source data are provided as a Source Data file.

elucidating the role of tumor necroptosis in tumorigenesis. Although it has been implied that RIPK1 may be involved in tumor necroptosis, the role of RIPK1 in tumor necroptosis has not been evaluated directly and the regulation of tumor necroptosis is largely unknown. In this current study, we found that RIPK1 is not required for tumor necroptosis during tumor development in MVT1 breast cancer model and may have an inhibitory effect on the induction of tumor necroptosis, this finding is consistent with previous reports that the necroptosis-inducing activity of ZBP1 is hindered by RIPK1 during normal embryonic development[43]. Importantly, we demonstrated that ZBP1 is the key mediator of tumor necroptosis during tumor development in both MVT-1 and B16 syngeneic orthotopic cancer models and that ZBP1 deletion inhibits tumor metastasis in MVT-1 model. While both RIPK1 and ZBP1 are capable of recruiting RIPK3 to mediate necroptosis, our results indicate that there is little redundancy among these two proteins in mediating tumor necroptosis during tumor development. As RIPK1 is involved in multiple aspects of cellular responses to inflammation and stresses[18,20], it is possible that RIPK1 may be involved in the development of certain types of tumors by regulating these pathways, but not necroptosis. Previous reports studied the possible involvement of necroptosis in tumor development and metastasis, using RIPK1 or RIPK3 KO cells/mice without directly examining tumor necroptosis.

ZBP1 is only expressed in certain normal tissues such as bone marrow and lymphoid tissues[26,44] and is known as a interferon stimulated gene (ISG)[26,45]. Given the fact that tumor cells and infiltrating immune cells produce IFNs during tumor development, it is likely that the elevation of ZBP1 expression in tumors is achieved by IFNs induction. We confirmed that ZBP1 expression could be induced by IFNs in MVT-1 tumor cells. Interestingly, while both RIPK3 and ZBP1 expression in MVT-1 cells are induced by the DNA methyl-transferase inhibitor, 5-AD, the induction of these two genes may be through different mechanism in response to 5-AD treatment as RIPK3 expression is only restored by 5-AD-induced DNA demethylation, but not IFNs (Supplementary Fig. 6a). The increase of ZBP1 expression by 5-AD is possibly triggered by the cellular stress responses to 5-AD treatment.

Although hypoxia and nutrient deprivation are thought to be the causes of tumor necrosis, our study showed that only GD induces ZBP1-dependent necroptosis in primary tumor cells. Interestingly, extended treatment for more than 3 days under GlnD or hypoxia did not result in any necroptosis either except some apoptosis in the primary cells (Supplementary Fig. 4d, e). Importantly, our finding that the glucose analog, 2DG, blocks GD-induced MLKL phosphorylation in tumors suggests that glucose deprivation plays a key role in tumor necroptosis during

tumor development. In addition, some recent publications showed that ZBP1 acts as a key mediator of IFNs-induced necroptotic cell death[31,46,47]. In these studies, IFNs trigger necroptosis in the presence of cell cycle inhibitors or IAP inhibitors[47]. As IFNs are known to induce apoptosis in tumor cells[48], we tested if IFNs induce necroptosis in primary MVT-1 and PyMT tumor cells and only observed minor apoptosis, but no necroptosis (Supplementary Fig. 4f, g). This observation is consistent with the fact that most tumor cells do not undergo necroptosis except those in the core regions of tumors although they are exposed to IFNs during tumor development. Therefore, our data suggests that IFNs is not the trigger of tumor necroptosis during tumor development. We tested the possibility that TNF signaling is involved in GD-induced apoptosis in ZBP1 KO cells. Neutralizing anti-TNFα antibody has no effect on GD-induced cell death in ZBP1 KO MVT-1 cells (Supplementary Fig. 6b). As GD leads to mitochondria damage, the remaining cell death is most likely due to the mitochondria-mediated intrinsic apoptotic pathway, which is in a slower kinetics comparing to necroptosis in WT cells.

We found that RIPK1 levels are dramatically decreased in tumor cells. While this decrease of RIPK1 level in tumor cells may accelerate ZBP1-induced tumor necroptosis, loss of RIPK1 combined with the increase of ZBP1 expression is not sufficient to trigger necroptosis in tumor cells as observed in RIPK1 KO MVT-1 tumors (Fig. 1). In other words, these RIPK1 KO cells still need some additional signal such as GD in the core region of tumors to undergo necroptosis. We found that GD triggers the release of mtDNA, which binds to ZBP1 and is essential for GD-induced necroptosis. The essentiality of Zα2 domain of ZBP1 in GD-induced necroptosis supports the idea that ZBP1 sense the cytosolic mtDNA to initiate tumor necroptosis. Interestingly, we found that the mitochondrial protein NOXA is crucial for GD-induced mtDNA release. Previous studies suggested that NOXA plays a key role in the process of mitochondrial membrane damage during apoptosis. Our finding suggests that NOXA may play a similar role in mediating the release of mtDNA during GD-induced necroptosis. However, because NOXA is known to be involved in multiple aspects of mitochondrial functions and biology, such as inhibiting MCL1 and BCL2 functions, and the level of NOXA expression is critical for tumor growth[49,50], and also tumor growth and mitochondrial functions are key factors in regulating the onset and the kinetics of tumor necrosis. In addition to mtDNA release, the role of NOXA in tumor necrosis will be complex and could not be evaluated by simply examining necrosis. The role of NOXA in tumor necrosis and necroptosis will be systematically studied in our future work.

While recent literature demonstrated the involvement of necroptosis in tumor growth, many fundamental questions

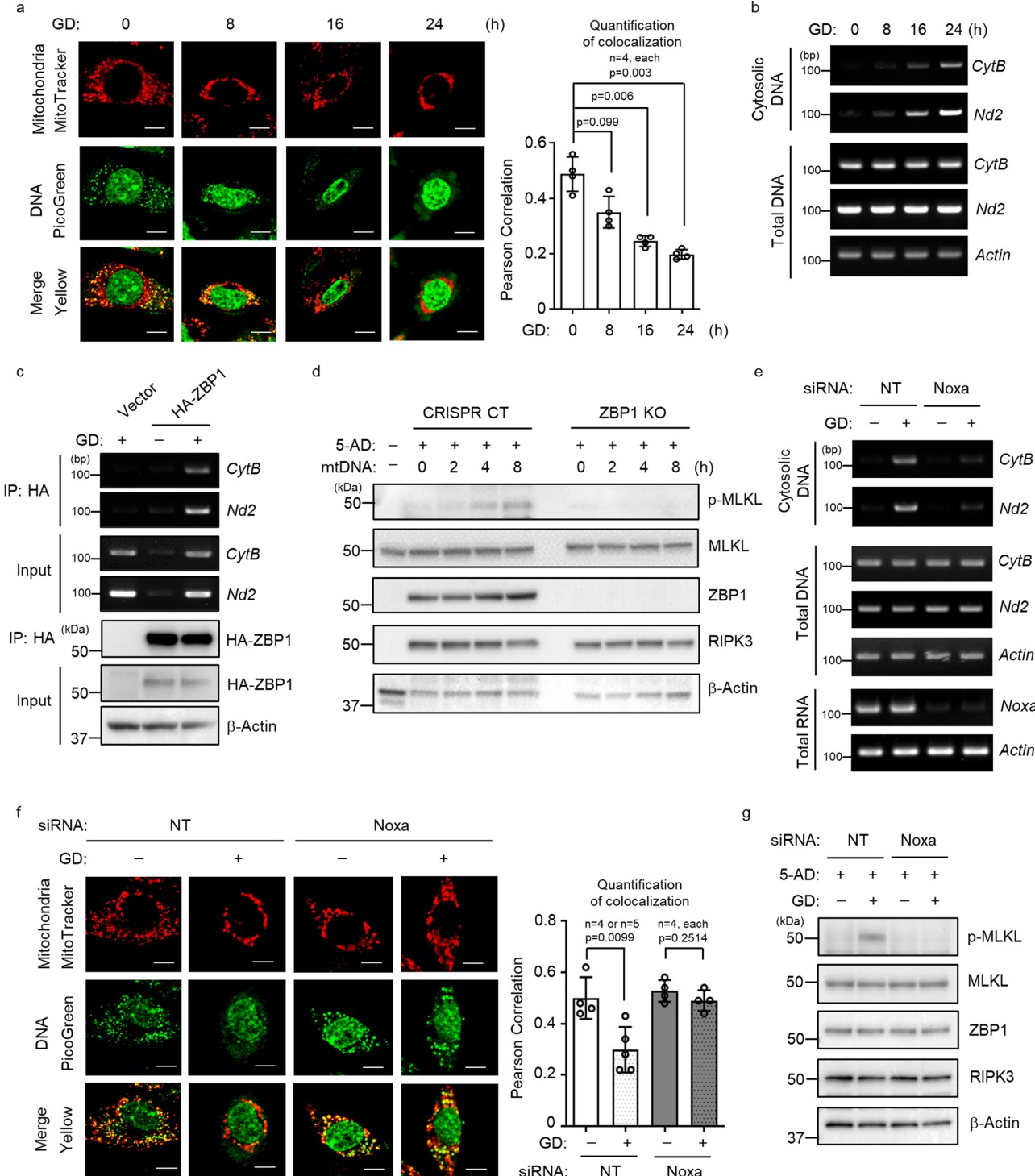

regarding the regulation and the role of necroptosis in tumorigenesis remain elusive. For example, we found that blocking tumor necroptosis significantly reduces tumor metastasis. However, we still do not know the underlying mechanism about how necroptosis promotes metastasis. Necroptosis in known to promote inflammation and we found that it is true that the macrophages isolated from necroptosis bearing WT tumors are more inflammatory comparing to those from necroptosis null tumors (Supplementary Fig. 6c). Future work is needed to evaluate the possible role of necroptosis-induced inflammation in metastasis. Nevertheless, our work reveals ZBP1, not RIPK1, as the key mediator upstream of RIPK3 in tumor necroptosis and provide

new insights about the regulation of tumor necroptosis during tumor development. Particularly, our work identified ZBP1 as a potential target for inhibiting tumor necroptosis as a possible cancer therapy.

## Methods

**Reagents and antibodies**. Necrostatin-1 (Nec-1) was purchased from Tocris bioscience. 5-AD (5-Aza-2′-deoxycytidine, A3656) and 2-Deoxy-D-glucose (2DG) was purchased from Sigma. Interferon β and γ were purchased from R&D systems. Anti-ZBP1 (AG-20B-0010) for mouse from AdipoGen; phospho-MLKL (ab196436) for mouse, MLKL (184718) and RIPK3 (ab72106) for human from Abcam; RIPK3 (2283) for mouse from ProSci; RIPK1 (610459) from BD Biosciences; Ki-67 (12202S), STING (13647), PUMA (14570S), DNMT1(5032), cleaved

**Fig. 6 Glucose deprivation (GD) promotes mitochondrial DNA release and triggers necroptosis. a** Confocal microscopy analysis of MVT-1 cells under GD condition and stained with MitoTracker (red) and PicoGreen (green) for DNA. (Scale bar, 5 μm; left panel). Quantification of colocalization of MitoTracker and PicoGreen staining using Image J software (right panel). Data are presented as mean values ± SEM. **b** Cytosolic fractions isolated from an equal number of MVT-1 cells under GD condition for the indicated time points were analyzed by PCR for mitochondrial *CytB* and *Nd2*. This PCR is representative of three independent experiments. **c** MVT-1 stably overexpressing HA-tagged ZBP1 were treated with GD. HA-ZBP1 was pulled down by Immunoprecipitation (IP) with anit-HA antibody from the cytosolic fraction, followed by PCR for mitochondrial *CytB* and *Nd2*. This PCR and IP are representative of three independent experiments. **d** MVT-1 cells were treated with 5-Aza-2'-deoxycytidine (5-AD), for 3 days and were then transfected with mitochondrial DNA (mtDNA, isolated from cytosol of cells under GD condition) for the indicated time and analyzed by western blotting using the indicated antibodies. This blot is representative of three independent experiments. **e** Cytosolic fractions isolated from an equal number of MVT-1 cells transfected with non-targeting siRNA (NT) or siRNA targeting Noxa (siNoxa), followed by GD condition, were analyzed by PCR for mitochondrial *CytB* and *Nd2*. This PCR is representative of three independent experiments. **f** Cells from Fig. **e** were stained with MitoTracker (red) and PicoGreen (green) as in Fig. **a** and analyzed by confocal microscopy (Scale bar, 5 μm; left panel). Quantification of colocalization of MitoTracker and PicoGreen staining was performed using Image J software (right panel, $n = 4$ or 5 biologically independent samples). Data are presented as mean values ± SEM. **g** MVT-1 cells were transfected with NT or siNoxa as in Fig. **e** and further treated with 5-AD for 3 days, followed by GD condition for 16 h and the lysates were then analyzed by western blotting using the indicated antibodies. This blot is representative of three independent experiments. Two-sided Student's *t* test was used to determine the statistical significance of differences between groups. Differences with *P* values < 0.05 were considered significant. Source data are provided as a Source Data file.

Caspase-3 (9664), Caspase-3 (14220) and HIF1-α (3434) from Cell Signaling Technology; ZBP1 (AF6309) for human from R&D systems; ZBP1 (sc-271483) from Santa Cruz Biotechnology; HA (sc-80s) from Santa Cruz Biotechnology; β-Actin (A3853) from Sigma.

**Mice.** Female FVB/NJ, and FVB/N-Tg (MMTV-PyVT) 634Mul/J (MMTV-PyMT) mice were purchased from The Jackson Laboratory. All animal experiments were performed under protocols approved by National Cancer Institute Animal Care and Use Committee and followed NIH guidelines. For orthotopic model, MVT-1 cells (derived from mammary tumor in MMTV-Myc-VEGF bitransgenic mouse were suspended in 100 μl of Matrigel Matrix (Corning) solution (diluted 1:1 with PBS) and then injected ($2 \times 10^6$/mouse) into the right inguinal mammary fat pad of FVB/NJ mice. Tumor volume was monitored weekly. Mice were routinely euthanized when tumors reached to 1500 mm$^3$ in volume and the whole tumor mass was excised. For B16 melanoma cells or LLC mouse lung cancer cells were suspended in 100 μl of Matrigel Matrix solution (diluted 1:1 with PBS) and then injected ($5 \times 10^6$/mouse) into the flank of C57BL/6J mice. For human breast cancer cells, MCF7 cells or MDA-MB-231 cells were suspended as described in above and injected ($5 \times 10^6$/mouse) into nude (BALB/c-nu/nu) mice. Tumor volume was monitored every week upon inoculation. For tail vein injection, the mice were injected with MVT-1 CRISPR CT or MVT-1 ZBP1 KO cells ($1 \times 10^6$/mouse) into tail vein. The lungs from each group were collected at 2 weeks. For Necrostatin-1 (Nec-1) injection, the mice were injected with MVT-1 cells into mammary fat pad and treated with *i.v.* injection of vehicle or Nec-1 (2.5 mg/kg, mixed with PBS) at 3-week until 5-week by weekly. For 2-Deoxy-D-glucose (2DG) injection, the mice were injected with MVT-1 cells into mammary fat pad and treated with *i.p.* injection of vehicle or 2DG (1 g/kg, mixed with PBS) at 3-week until 5-week by daily. To be comparable to the late stage necrotic MMTV-PyMT and MVT-1 tumors, the tumors from these models were collected as indicated.

**RNA-seq analysis of TCGA data.** RNA-Seq data from The Cancer Genome Atlas (TCGA) Research Network was retrieved from NCI's Genomic Data Commons (GDC) repository using the TCGABiolinks R package[51]. Data from GDC's harmonization pipeline[52] were downloaded as upper-quantile normalized fragments-per-kilobase-exon (FPKM) values for 56,512 transcripts in each of 11,093 samples from 33 TCGA datasets, along with all available phenotype information. For visualization purposes, only the 15 datasets that contained more than ten normal samples were included. Summarization and visualization were performed in R version 3.6.0[53] using the dplyr (version 0.8.3) and ggplot2 (version 3.2.1) packages. For the breast cancer dataset (TCGA-BRCA), differential expression analysis was performed for each tumor stage separately. Lowly expressed genes with a mean FPKM of less than 100 were removed, yielding 46,172 transcripts from 1190 samples. FPKM values were log2-transformed and differential expression was computed using the 'limma' package (version 3.40.6)[54], allowing for the prior variance to be adjusted according to the intensity-variance trend. *P* values were adjusted for using the Benjamini–Hochberg procedure.

**Primary cell culture.** We modified and used previously reported methods to isolate tumor cells from primary tumors[55,56]. Briefly, MMTV-PyMT tumors from 15-week-old MMTV-PyMT mice or MVT-1 tumors from mice injected with MVT-1 cells at 5 weeks were collected. The tumor tissue was digested in PBS buffer containing 0.5% BSA (Sigma), 2 mM EDTA (Quality Biological), collagenase II (Sigma), collagenase IV (Sigma) and DNase I (Sigma) for 15 min at 37 °C. After incubation, the cells were filtered with 100 μm cell strainer (Corning) and were

added with ACK lysing buffer (Quality Biological) to remove RBCs and plated into 10 cm dish. After 16 h, the cells were replated as needed for further experiments.

**GFP-MVT-1 sorting.** MVT-1 cells were lentiviral transfected with CMV-GFP-T2A-Luciferase Lentivector (SBI system biosciences), successfully transfected MVT-1 cells were GFP positively sorted by using BD FACSAria Fusion sorter (BD bioscience). Then, GFP-MVT-1 cells ($2 \times 10^6$) were injected into fat pad of FVB/NJ mice. After 5 weeks, tumor was collected and digested in PBS buffer as in above described. The GFP-positive MVT-1 cells were then sorted from total cells by using a BD FACSAria Fusion sorter. After sorting, GFP-positive MVT-1 cells were lysised in M2 buffer and used for further Immunoblot analysis.

**Flow cytometry.** A flow cytometer (Sony SA3800) was used for analyses. Relative change in fluorescence was analyzed with FlowJo software. For measurement of cell proliferation, cells were labeled with FITC BrdU Flow Kit (51-2354AK, BD Pharmingen) according to the manufacturer's protocol. For analysis of cell death, cells were labeled with propidium iodide (PI, Invitrogen).

**Histology analysis.** Tumors were bisected into two pieces in the middle of the tumor at the longest diameter orientation using a razor blade: one-half of the tumor was immediately placed in 4% buffered formalin (Z-fix) overnight, and the other half was frozen for protein extraction. The fixed tumors were embedded in paraffin and cut into 5 μm-thick serial sections staining using standard histological procedures. Every 3rd slide was routinely stained with haematoxylin and eosin as described[57]. Tumor sections from the center of excised tumors of similar size were used for analysis. Tumor necrosis was designated on H&E-stained slides as areas of dark-haematoxylin-stained necrotic tumor cells immediately adjacent to light-haematoxylin-stained viable tissues[58]. The quantitation of tumor necrotic/death area was counted using Image J and represented as the percentage of tumor necrotic/death area within whole tumor. Mice tumor sections (4 μm) were deparaffinized by incubation at 56 °C for 30 min and subsequent xylene washes then rehydrated with a graded ethanol. The paraffin sections were subjected to antigen retrieval with retrieval buffer (Dako) at 95 °C for 10 min and cooled down until room temperature. The slides were then treated with 3% $H_2O_2$ for 5 min washed with phosphate-buffered saline (PBS). The slides then were blocked with 2% normal goat serum, followed by overnight incubation with primary antibodies against p-MLKL (1:5000) or cl.Casp-3 (1:1000). Signals were developed using VECTASTIN ABC Elite kit (Vector Laboratories) and DAB Substrate Kit (Vector Laboratories) followed by manufacturer's instructions. For in situ detection of apoptosis, the slides were stained with TUNEL kit (ab206386, abcam) according to the manufacturer's protocol. The slides were counter stained with Hematoxylin (Vector Laboratories) to detect nucleus. For the quantitation of metastatic burden, paraffin-embedded lung tissues were sectioned 400 μm apart throughout the whole lung followed by H&E staining. The frequency of the metastatic foci was counted manually in a blinded fashion.

**Generation of ZBP1 knockout cells by CRISPR.** MVT-1 cells were maintained in DMEM containing 10% FBS and 1% penicillin/streptomycin mixture. To generate ZBP1 knockout (ZBP1 KO) MVT-1 cells, lentiviral sgRNA vector targeting ZBP1 was constructed by ligation of hybridized oligos into LentiCRISPR V2 (pXPR_001, GeCKO) vector. ZBP1 gRNA sequences are shown in Supplementary Table 1. 293T cells were transfected with sgRNA vector, pPAX2 (Addgene) and pCMV-VSV-g (Addgene) for 24 hr. MVT-1 cells were then infected with lentivirus-containing supernatants with polybrene (Millipore) for 24 h. After 3 days of selection with puromycin (2 μg ml$^{-1}$, Sigma-Aldrich), the ZBP1 KO cells were

placed into 96 well plate to undergo clonal selection without puromycin. Single clones with complete knockout of ZBP1 was verified by DNA sequencing and western blot analysis.

**Generation of RIPK1 knockout cells by inducible CRISPR system**. MVT-1 inducible RIPK1 KO stable cells were generated using the Lenti-X tet-on 3G CRISPR-Cas9 system from Clontech (Takara Bio). Briefly, 293T cells were transfected with pLVX-EF1a-Tet3G, pLVX-TRE3G-Cas9-puro, and pLVX-hyg-sgRIPK1 for 48 h. RIPK1 gRNA sequences are shown in Supplementary Table 1. MVT-1 cells were sequentially transduced with lentivirus-containing supernatants with polybrene and selected with G418 (500 µg ml$^{-1}$), puromycin (2 µg ml$^{-1}$), and hygromycin (1 mg ml$^{-1}$), respectively. The stable cells were treated with 2 µg ml$^{-1}$ doxycycline for two weeks before clonal selection. Single clones with complete knockout of RIPK1 was verified by western blot analysis.

**Western blotting analysis and immunoprecipitation**. Tumor tissues or in vitro cultured cells were lysed in RIPA buffer or M2 buffer, respectively. Tumor and cell lysates were separated by 4–20% SDS-polyacrylamide gel electrophoresis, followed by probing with anti-MLKL, anti-mouse p-MLKL, anti-RIPK1, anti-RIPK3, anti-ZBP1, anti-cl.Casp-3, anti-HA, or anti-β-Actin antibodies. Signals were developed by using enhanced chemiluminescence kit (Bio-Rad). For immunoprecipitation, the lysates were precipitated with antibodies (1 µg) and protein-G agarose bead by incubation at 4 °C overnight. The beads were washed four to six times with 1 ml of M2 buffer and the bound proteins were removed by boiling in SDS buffer and resolved in 4-20% SDS-polyacrylamide gels for western blot analysis.

**Transwell migration assay**. To assess tumor cell migration activity, $2 \times 10^4$ MVT-1 CRISPR-CT or MVT-1-ZBP1 KO cells were cultured in serum-free DMEM in the upper compartment of a transwell insert (8 µm pore size, Corning). The lower compartment was filled with DMEM containing 10% FBS. After 12 h, cells in the upper compartment were removed and the transwell membrane was stained with 0.5% crystal violet (Sigma-Aldrich). Migrated cells on the membrane were counted under a light microscope (Zeiss).

**Quantitative RT-PCR**. Tumor lysates or in vitro cultured cells were subjected to RNA extraction using TriPure isolation reagent (Roche) along with a chloroform extraction method. cDNA synthesis was conducted using Superscript III First-strand synthesis kit (Invitrogen). Predesigned qPCR primers for ZBP1 and β-Actin (Integrated DNA Technologies) were used and relative mRNA expression was measured using SensiFAST Probe Hi-ROX Mix (Bioline) on QuantStudio 3 Real-Time PCR System (Thermo Fisher Scientific). $2^{-\Delta\Delta CT}$ method was used to quantify fold induction. To determine RIPK1 and RIPK3 (primer, qPCR was performed with SYBR Green Master Mix (Applied Biosystems) on an ABI StepOnePlus system according to the manufacturer's protocol. Each cDNA data was normalized by β-Actin expression. Primer sequences are shown in Supplementary Table 1.

**Methylation-specific PCR**. Methylation assay were performed as previously described[27]. Briefly, Genomic DNA was extracted using Quick-DNA Miniprep kit (ZYMO Research, D3024). Methylation status of RIPK3 was determined by methylation-specific PCR. Briefly, genomic DNA was bisulphite-treated with BisuFlash DNA Bisulfite Conversion Easy Kit (Epigentek, P-1054-050). Bisulphite-treated DNA was amplified using primers specific for either methylated or unmethylated DNA. The sequences of methylated-specific (M) primer and unmethylated-specific (U) primer for RIPK3 are shown in Supplementary Table 1.

**Plasmid construction**. Mouse ZBP1 vector was obtained from Vector Builder. Mouse ZBP1 point mutant for RHIM A domain and Zα2 domain was generated by using site-direct mutagenesis. Briefly, to generate RHIM A domain point mutant, four amino acids IQIG from 192 to 195 of ZBP1 were substituted to four alanine AAAA and confirmed by DNA sequencing. To generate Zα2 domain point mutant, asparagine from 122 and tyrosine from 126 were substituted to aspartate and alanine, respectively, and confirmed by DNA sequencing.

**shRNA**. The shRNA lentiviral plasmids were purchased from Sigma. The shRNA against mouse ZBP1 (NM_021394) or Noxa (NM_021451.1) corresponds to the 3′-untranslated region. The genomic DNA was isolated for PCR.

**siRNA**. siRNAs (SMARTpool) against mouse Noxa, human NOXA and non-targeting siRNA were purchased from Dharmacon and transfected using RNAi lipofectamine (Invitrogen) according to manufacture protocol. To determine the knockdown effect of Noxa, we performed and confirmed by PCR. The sequences of primers for mouse Noxa and human NOXA are shown in Supplementary Table 1.

**Mitochondrial DNA analysis**. Cytosolic fraction was isolated using Qproteome Mitochondria isolation kit (Qiagen) according to the manufacturer's protocol.

DNA was extracted from cytosolic fractions, and total cell lysates using the Quick-DNA MiniPrep Kit (Zymo Research) according to the manufacturer's protocol. Presence of mitochondrial DNA was confirmed by PCR. The sequences of primers for mitochondrial genes are shown in Supplementary Table 1.

**Confocal fluorescence microscopy**. MVT-1 cells were treated with GD. Primary MVT-1 cells isolated from 5-week mice were implanted to µ-Dish ibiTreat (80136, Ibidi) and treated for 16 h with GD. Mitochondria and mitochondrial DNA were analyzed by confocal fluorescence microscopy after cell staining with PicoGreen (Thermo Fisher Scientific) for 1 h at 37 °C and 25 nM MitoTracker Red CMXRos (Thermo Fisher Scientific) for 10 min. Colocalization of MitoTracker Red and PicoGreen was quantified by analysis of Pearson's correlation using Image J software (https://imagej.nih.gov/ij/).

**Super-resolution microscopy**. Images were acquired using a Nikon Ti-2 microscope equipped with a Yokogawa CSU-W1 SoRa spinning disk unit, 20x plan-apochromat (N.A. 0.75) and 60x plan-apochromat (N.A. 1.49) objective lenses, and Photometrics BSI sCMOS camera. Confocal extended field of view tile images were collected using the CSU-W1 confocal spinning disk and 20x objective lens, and stitched together using the Nikon Elements software. Super-resolution images were collected using the 60x objective and the SoRa spinning disk. Z-stacks were collected using a 0.200 um step size. The images were deconvolved using a Richardson-Lucy constrained iterative algorithm included in the Nikon Elements software.

**mtDNA FISH**. Fluorescence in situ hybridization (FISH) of mtDNA on formalin-fixed, paraffin-embedded (FFPE) tissues[36] was performed manually followed by manufacturer's instructions using ViewRNA™ ISH Tissue Assay kit (Thermo Fisher, 19931). Briefly, the FFPE slides were baked at 60 °C for 1 h, and subsequent xylene washes 3 times for 5 min. Then the slides were rehydrarted twice in 100% ethanol for 5 min. The slides were subjected to 1X Pretreatment Solution at 95 °C for 15 min and incubated with protease solution at 40 °C for 30 min. The slide then were fixed with 10% NBP (4% formaldehyde in PBS) for 5 min at room temperature. The slide were incubated with Mus musculus Cyb561 (1:40, Thermo Fisher, VB1-3029378) at 40 °C for 2 h followed by the standard amplification steps as the manufacturer's instruction. The slides were counter stained with IraZolve-Mito (Cayman Chemical, 25910)[37] to detect mitochondria.

**Statistical analysis**. All data were analyzed with the Graphpad Prism 8 software. Two-sided student's $t$ test was used to determine the statistical significance of differences between groups. Differences with $P$ values < 0.05 were considered significant.

**Reporting summary**. Further information on research design is available in the Nature Research Reporting Summary linked to this article.

## Data availability

Data supporting the findings of this study are available within the article, its Supplementary Information files, and from the corresponding author upon reasonable request. A reporting summary for this article is available as a Supplementary Information file. TCGA data referenced in this study are available from NCI's Genomic Data Commons (GDC) repository, and were accessed using the R package 'TCGABiolinks'. Source data are provided with this paper.

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

## Acknowledgements

This research was supported by the Intramural Research Program of the Center for Cancer Research, National Cancer Institute, National Institutes of Health.

## Author contributions

J.Y.B. designed and performed most of the experiments. Z.S.L. conducted some mouse experiments, and mitochondrial experiments. D.J. and H.K. conducted some mouse experiments. J.Y. generated RIPK1 knockout cells. C.K., M.C., and Z.C. helped with experiments. R.L. and M.K. helped with confocal microscopy. M.T. helped with TCGA data analysis. S.C. helped with experiments and manuscript preparation. Z.-G..L. conceived, supervised and directed the project and wrote the manuscript.

## Funding

## Competing interests

The authors declare no competing interests.
