## [Peer Review File · Nature Communications]

REVIEWER COMMENTS

Reviewer #1 (Remarks to the Author):

There have been a few reports that necroptosis plays a role in tumour development (e.g. Seifert, Nature) but in these cases necroptosis was occurring in cells in the tumour micro-environment and not in the malignant cells themselves. Of note, this paper and related ones have been seriously questioned for example in Patel et al CDD 2020. This work has not been taken into consideration by the authors of this manuscript.

In this paper JY Baik and colleagues present data that led them to conclude that necroptosis in the malignant cells reduces tumour growth and metastasis. They go on to show that ZBP1, but not RIPK1 (which is critical for TNF induced necroptosis), is essential for the necroptosis that occurs in growing tumours. They also show that this necroptosis is triggered by the deprivation of glucose. The major problem with the work presented is that the most important experiments, demonstrating areas of tumour necrosis (e.g. Figures 1, 3, 4, 5), are of insufficient technical quality to allow conclusions to be drawn. These data are all based on a single section in one orientation. Depending on how a tissue is cut, very different results may appear. Reliable data on the area of necrosis in a tumour requires serial sectioning and 3D reconstruction to generate reliable measurements of areas of necrosis. In the absence of such data, the conclusions are not supported.

Moreover, it is not clear to me how many independent ZBP1 knockout lines/clones for each tumour had been generated and examined independently. To allow firm conclusions, at least three independent lines/clones should have been generated, preferably using three independent sgRNAs to target ZBP1.

The data on the release of mitochondrial DNA into the cytosol (Figure 6) are not highly convincing. The authors should present image analysis, as presented in K McCarthy et al, Science demonstrating release of DNA from mitochondria.

Reviewer #2 (Remarks to the Author):

Baik et al. examined the mechanism of tumor necrosis in cancers. The authors found that tumor necrosis was not inhibited by RIPK1 inhibitor Nec-1, nor did deletion of RIPK1 itself affect the tumor necrosis. The authors found that the levels of Z-DNA-binding protein 1 (ZBP1), and RIPK3, but not RIPK1, were increased during tumor development in preclinical cancer models. The authors further showed that knockout of ZBP1 promotes apoptosis and inhibits necroptosis in tumor cells. Deletion of ZBP1 blocks tumor necroptosis during tumor development and inhibits metastasis. The authors further showed that glucose deprivation triggers ZBP1-dependent necroptosis in tumor cells. Glucose deprivation causes mitochondrial DNA (mtDNA) release to the cytoplasm and the binding of mtDNA to ZBP1 to activate MLKL in a NOXA-dependent manner. The authors conclude that ZBP1 as the key regulator of tumor necroptosis and provides a potential drug target for controlling tumor metastasis.

Induction of ZBP1 during tumor development is quite robust. The possible role of ZBP1 activated by binding to mtDNA in tumors is interesting. I just have some specific questions:

- 1) Since RIPK3 expression is often silenced in tumors as reported by the authors previously, would ZBP-1 still be induced without RIPK3 to activate MLKL and necroptosis in native tumors when RIPK3 levels are low?
- 2) Can 5-AD treatment contribute to the release of mtDNA that can in turn bind to ZBP-1?
- 3) Since ZBP-1 is also induced in xenograft tumors to mediate necrosis, is mtDNA also known to be released in these tumor models?

Reviewer #3 (Remarks to the Author):

Baik and colleagues provide compelling data that ZBP1-mediated necroptosis underlies tumor necrosis in vivo and tumor cell line colonization to lung. They show that in the tumor cell lines tested ZBP1 and RIPK3 are over expressed but that RIPK1 is not required. Interestingly they provide evidence that ZBP1 activation requires nucleic acid sensing and that glucose deprivation triggers mito DNA release which activates the ZBP1-RIPK3-MLKL pathway. Perhaps most interesting is that ZBP1 deletion in MVT mammary tumor cell line reduces lung colonization but the reasons for this are unclear. Is this dependent on the Z-DNA binding domain? Can this be shown in B16 or other tumor cell line 'metastasis' models? Does the inhibition of necroptosis reduce inflammatory cytokines/chemokines that drive seeding into the lung?

Overall these findings are novel and interesting with major implications for cancer research and therefore appropriate for publication in Nat Comm.

Major comments:

1. The authors show that ZBP1 deletion reduces tumor necrosis but results in increased apoptosis. Is this IFN mediated or RIPK1-dependent apoptosis?
2. In several figures GD (or ZBP1 deletion) results in MLKL phosphorylation and RIPK3 degradation (Figs 4a,b;5b;Ext data Fig4a,b). This is unexpected as RIPK3 is required for MLKL phosphorylation.
3. The authors treat with 5-azacytidine to increase RIPK3 and ZBP1 levels in the mammary tumor cell lines but this treatment has broad effects. Can IFNg be used instead to at least increase ZBP1 levels? This may be sufficient to activate the pathway upon GD.
4. In Fig 4b, ZBP1 ko partially rescues tumor cell death upon GD and they show that STING does not contribute. Can the cell death be blocked with anti-TNF abs?
5. The data that mito DNA binds ZBP1 is suggestive but not convincing based on assays used and the ZBP1 overexpression. Since the ZBP1 abs are limiting, can CRISPR be used to knock in a tag to the endogenous locus and IP done perhaps in IFNg treated MVT cells?

Minor comments:

1. The ms could benefit from additional editing and to avoid overinterpretation. The data should be reported based on the mammary or other tumor cell line examined rather than making broad statements such as "RIPK1 is not required for tumor cells to undergo necroptosis during tumor development." line 93-4.
2. line 100: spontaneous is not the correct term as this are GEMM
3. line 123: specify MVT mammary tumor cell line or model rather than tumor development
4. Does the expression of VEGF in MVT cell line influence the hypoxia assays or lung colonization? Can the effects of blocking necroptosis on metastasis be demonstrated in the B16 model?
5. line 234 and elsewhere: rather than following GD treatment it would be better clearer to state under conditions of GD.

Pinot-to-point response:

Reviewer # 1

There have been a few reports that necroptosis plays a role in tumour development (e.g. Seifert, Nature) but in these cases necroptosis was occurring in cells in the tumour micro-environment and not in the malignant cells themselves. Of note, this paper and related ones have been seriously questioned for example in Patel et al CDD 2020. This work has not been taken into consideration by the authors of this manuscript.

Response: We thank the reviewer for pointing out that we missed the important work by Patel et al in our manuscript. As the reviewer indicated that while Dr. Miller's group (Siefert, Nature) suggested that the RIPK1/RIPK3 necroptotic pathway may be involved in pancreatic oncogenesis, Dr. Vucic's group (Patel et al CDD 2020) showed that RIPK1 is needed for inflammatory diseases, but not tumor growth or metastasis. We apologize for not discussing this important work on RIPK1 in tumorigenesis. In our revised manuscript, we discussed this work in the introduction and discussion sections. We thank the review for the information as this work is consistent with our finding that RIPK1 is not involved in tumor necroptosis. While these reports studied the possible involvement of necroptosis in tumorigenesis, they mainly used RIPK1 or RIPK3 KO cells/mice without directly examining tumor necroptosis. Actually, we are the first one to show that necroptosis happens in tumor cells in tumor necrotic areas (Cell Research 2018).

In this paper JY Baik and colleagues present data that led them to conclude that necroptosis in the malignant cells reduces tumour growth and metastasis. They go on to show that ZBP1, but not RIPK1 (which is critical for TNF induced necroptosis), is essential for the necroptosis that occurs in growing tumours. They also show that this necroptosis is triggered by the deprivation of glucose.

The major problem with the work presented is that the most important experiments, demonstrating areas of tumour necrosis (e.g. Figures 1, 3, 4, 5), are of insufficient technical quality to allow conclusions to be drawn. These data are all based on a single section in one orientation. Depending on how a tissue is cut, very different results may appear. Reliable data on the area of necrosis in a tumour requires serial sectioning and 3D reconstruction to generate reliable measurements of areas of necrosis. In the absence of such data, the conclusions are not supported.

Response: We thank the reviewer for raising this critical issue regarding tumor necrosis as we did not address it clearly in our manuscript (Fig. 1, 3). As the reviewer pointed out, because tumors grow in different shapes, the orientation of tumor tissue cutting could result in very different size of tumor necrosis in a single section. However, in the case of MVT-1 tumors, when MVT-1 cells are properly (not too deep or too shallow) injected into fat pads of mice with matrigel, they normally grow to ball-shaped tumors. When tumors reached about 1500 mm³ in volume, mice were euthanized and the whole ball-shaped tumor mass was excised. We regularly cut the tumors into two pieces in the middle of the tumor at the longest diameter orientation. Tumor serial sections (5 sections, the middle one for H&E) from the centers of excised similar sized tumors as shown in our figures were used for analysis and quantification (n=4-7). As tumor necrosis normally happens at the center areas of tumors, those center sections of tumors mostly display the largest areas of tumor necrosis in MVT-1 tumors. We followed the reviewer's suggestion and performed 500 μm-interval serial sections of a pair of similar size WT and ZBP1

KO MVT-1 tumors. After the first 1/3 of the tissue was removed from one side of the tumor, we cut a set of 3 serial sections (5 μm -thick, with the 2nd slide stained with H&E) for 12 intervals every 500 μm through the remaining tissue. As shown below, the center sections of these tumors represent the largest areas of tumor necrosis. Therefore, these serial section results further suggest that, as these MVT-1 tumors are ball-shaped, orientation is less of an issue for the bias of different tumor sections in most cases of MVT-1 tumors. But nevertheless, as orientation of tissue cutting is indeed a critical issue in studies with tumor tissue sections, we routinely used the center sections of tumors at the longest diameter orientation in our study on tumor necrosis and quantifying the necrotic areas with multiple tumors at the similar size. The reviewer suggested to build a 3-D image to examine necrosis but as there is no specific marker for necrosis, we used DAPI to stain viable cells in the 12 serial sections of tumor as described above and build a 3-D image. We used confocal extended field of view tiled images collected using the CSU-W1 confocal spinning disk and 20x objective lens, and stitched together using the Nikon Elements software. However, the DAPI staining/3-D imaging did not provide any additional information than that from the H&E staining. Therefore, we did not include this DAPI staining/3-D image in our revised manuscript.

To examine the status of necroptosis in different tumors, we checked MLKL phosphorylation by both IHC of tissue sections of one half of each tumor and Western blotting with the other half of the tumor (Fig. 1, 3, 4, 5). The IHC staining and Western blotting results of MLKL phosphorylation are highly consistent for most tumors examined. The Western blotting results further supported that our IHC study on necroptosis is not biased. Also, in IHC study of MLKL phosphorylation, H&E staining of the serial sections showed dead areas of tumors, but MLKL phosphorylation is negative in the same areas, which become cl. Casp-3 or TUNEL positive. Orientation of tissue cutting will not be a factor in these studies although we used the serial center sections of tumors at the longest diameter orientation in our studies. In our previous study with MLKL KO MVT-1 cells, we made the same observation that blocking necroptosis in tumor cells reduces tumor necrosis (Cell Research, 2018). In our revised manuscript, we added the information about using the center sections of tumors at the longest diameter orientation, examining MLKL phosphorylation by both IHC with tissue section of one half of each tumor and Western blotting with the other half of the tumor, and the below serial section data in our revised manuscript (Supplementary Fig. 3d).

Moreover, it is not clear to me how many independent ZBP1 knockout lines/clones for each tumour had been generated and examined independently. To allow firm conclusions, at least three independent lines/clones should have been generated, preferably using three independent sgRNAs to target ZBP1.

Response: We apologize for not making this clear in our original manuscript. The data present in MVT-1 tumors are from two individual clones of the same sgRNA. The data in B16 tumors is from a shRNA pool cells. We did have data with the second sgRNA pool cells and a shRNA pool cells in MVT-1 tumors, but these data were not included in the manuscript. In our revised manuscript, we included the data from the second sgRNA pool cells (Supplementary Fig. 3e, f and l). As the shRNA is less effective to delete ZBP1 in MVT-1 cells comparing to sgRNAs, we did not include the shRNA data of MVT-1 cells in the revised manuscript. In addition, we have generated ZBP1 KO mice in MMTV-PyMT model and we observed the similar effect of ZBP1 deletion on tumor necrosis/necroptosis, tumor growth and lung metastasis. However, we are currently generating MMTV-ZBP1 conditional KO mice to confirm that ZBP1 deletion in tumor cells is responsible for the changes in this GEMM tumor model. Therefore, we did not include this MMTV-PyMT model data in the revised manuscript.

The data on the release of mitochondrial DNA into the cytosol (Figure 6) are not highly convincing. The authors should present image analysis, as presented in K McCarthy et al, Science demonstrating release of DNA from mitochondria.

Response: While we are following previous publications to examine the release of mtDNA into the cytosol, we agree with the reviewer that the approach published by McArthur et al (**K McArthur et al, Science, 2018**) using Live-cell Lattice Light-sheet Microscopy (LLSM) technique provides super resolution to reveal the mtDNA release from mitochondria. Following the approach described by McArthur et al, MVT-1 cells were co-transfected with vectors targeting mtDNA (peGFP-TFAM, green) or mitochondria (pLV-mitoDsRed, red). 48 hours after the transfection, the cells were treated with GD for the indicated time. Super-resolution images were collected using the 60x objective and the SoRa spinning disk. Z-stacks were collected using a 0.2 μm step size. The images were deconvolved using a Richardson-Lucy constrained iterative

algorithm included in the Nikon Elements software. As shown in the below images, some mtDNA (green) is clearly released from mitochondria after 16 hr GD treatment. We included this new data in our revised manuscript (Supplementary Fig. 5c).

Reviewer #2:

Baik et al. examined the mechanism of tumor necrosis in cancers. The authors found that tumor necrosis was not inhibited by RIPK1 inhibitor Nec-1, nor did deletion of RIPK1 itself affect the tumor necrosis. The authors found that the levels of Z-DNA-binding protein 1 (ZBP1), and RIPK3, but not RIPK1, were increased during tumor development in preclinical cancer models. The authors further showed that knockout of ZBP1 promotes apoptosis and inhibits necroptosis in tumor cells. Deletion of ZBP1 blocks tumor necroptosis during tumor development and inhibits metastasis. The authors further showed that glucose deprivation triggers ZBP1-dependent necroptosis in tumor cells. Glucose deprivation causes mitochondrial DNA (mtDNA) release to the cytoplasm and the binding of mtDNA to ZBP1 to activate MLKL in a NOXA-dependent manner. The authors conclude that ZBP1 as the key regulator of tumor necroptosis and provides a potential drug target for controlling tumor metastasis.

Induction of ZBP1 during tumor development is quite robust. The possible role of ZBP1 activated by binding to mtDNA in tumors is interesting. I just have some specific questions:

1) Since RIPK3 expression is often silenced in tumors as reported by the authors previously, would ZBP-1 still be induced without RIPK3 to activate MLKL and necroptosis in native tumors when RIPK3 levels are low?

Response: As the reviewer pointed out, it has been reported that RIPK3 expression is low in many types of tumors. Since RIPK3 is known to be essential for necroptosis, it is critical to address how ZBP1 induces necroptosis in RIPK3 low tumors. Without RIPK3, ZBP1 will unlikely induce tumor necroptosis. However, as we showed in our current study, while RIPK3 levels are very low/undetectable in the early stages of tumor development, RIPK3 is dramatically upregulated when tumor reaches certain stages in both MVT-1 and MMTV-PyMT tumors. This is also true for B16 and MCF7 tumors. We also showed in our previous study (Cell research, 2018) that RIPK3 levels are significantly increased in human necrotic breast cancers comparing to non-necrotic tumors. Therefore, it is likely that RIPK3 expression will be reprogrammed in most of solid tumors during tumor development when the tumor reaches a certain size, by then, the tumor cells in the center of tumors will be under hypoxia and nutrients deprivation. In other words, while RIPK3 expression is low in many solid tumors, its levels may be significantly elevated in later stages. Our new data showed below, suggest that this reprogramming of RIPK3

expression most likely happens by the reduction of DNA methylation of RIPK3. We included these new data in our revised manuscript (Fig. 2h, Supplementary Fig. 2i).

RIPK3 expression is increased in necroptotic tumors a, Methylation specific PCR of genomic DNA from MVT-1 or 5-AD treated MVT-1 or MVT-1 tumor lysates was detected by using methylation primer (M) or unmethylation primer (UM) or regular primer (R) for RIPK3. **b**, Western blotting analysis of MVT-1 cell lysates or MVT-1 tumor cell lysates collected from 5-week tumors of FVB/NJ mice implanted with MVT-1 cells using the indicated antibodies.

2) Can 5-AD treatment contribute to the release of mtDNA that can in turn bind to ZBP-1?

Response: We carried out ZBP1 IP experiments to examine if 5-AD treatment contributes to the release of mtDNA and the binding of mtDNA to ZBP1. As shown below, 5-AD treatment alone does not induce the release of mtDNA to the cytosol (input) and mtDNA binding to ZBP1 (IP) while GD alone is sufficient to trigger the release and binding. However, the presence of 5-AD made the cells more sensitive to GD and the release of mtDNA as shown (input), likely the toxicity of 5-AD made cells more fragile. We added these new data in our revised manuscript (Supplementary Fig. 5e).

MVT-1 stably overexpressing HA-tagged ZBP1 were treated with 5-AD for 3 days, then further treated with GD. HA-ZBP1 was pulled down by IP with anti-HA antibody from the cytosolic fraction, followed by PCR for detecting mitochondrial CytB and Nd2 (left panel). Right panel shows the pull-down efficiency of anti-HA IP as a control for left panel under the treatment.

3) Since ZBP-1 is also induced in xenograft tumors to mediate necrosis, is mtDNA also known to be released in these tumor models?

Response: While mitochondrial damage has been widely reported during tumorigenesis, the release of mtDNA during tumor development has been specifically investigated. To explore the possibility that mtDNA is released in the MVT-1 tumor model, we performed In Situ hybridization (ISH) imaging study with 2-week (non-necrotic) and 4-week (necrotic) tumor sections. In this experiment, mitochondria are labeled with IraZolve-Mito (green) and mtDNA/mtRNA are labeled with a Cyb561 RNA probe conjugated with Alexa Fluor 546 (red)^{1,2}. As shown in the below results, mitochondria and mtDNA/mtRNA are nicely co-localized in 2-week tumor cells, but mtDNA/mtRNA are clearly dissociated with mitochondria in some tumor cells of 4-week tumor. This preliminary data suggests that mtDNA may be released to the cytosol in xenograft tumors that bear necroptosis. We have included this data in our revised manuscript (Supplementary Fig. 5d).

MVT-1 tumor paraffin sections of 2-week or 4-week from mice implanted with MVT-1 cells were co-stained with mitochondria (IraZolve-Mito, green) and Cyb561 (RNA probe conjugated with Alexa Fluor 546, red) followed by manufacturer's procedure (ViewRNA ISH Tissue assay). Scale bar, 20 μ m

Reviewer #3:

Baik and colleagues provide compelling data that ZBP1-mediated necroptosis underlies tumor necrosis in vivo and tumor cell line colonization to lung. They show that in the tumor cell lines tested ZBP1 and RIPK3 are over expressed but that RIPK1 is not required. Interestingly they provide evidence that ZBP1 activation requires nucleic acid sensing and that glucose deprivation triggers mito DNA release which activates the ZBP1-RIPK3-MLKL pathway. Perhaps most interesting is that ZBP1 deletion in MVT mammary tumor cell line reduces lung colonization but the reasons for this are unclear. Is this dependent on the Z-DNA binding domain? Can this be shown in B16 or other tumor cell line ‘metastasis’ models? Does the inhibition of necroptosis reduce inflammatory cytokines/chemokines that drive seeding into the lung?

Overall these findings are novel and interesting with major implications for cancer research and therefore appropriate for publication in Nat Comm.

We thank Dr. Kelliher for concluding that our work is “*novel and interesting with major implications for cancer research and therefore appropriate for publication in Nat Comm*”. Regarding the role of the Z-DNA binding domain of ZBP1 in lung metastasis, we examined the lung metastasis in mice transplanted with WT or $Z\alpha 2$ mutant ZBP1 reconstituted ZBP1 KO MVT-1 cells. As shown below, while WT ZBP1 restored the metastasis of ZBP1 KO cells, $Z\alpha 2$ mutant ZBP1 failed to do so. Therefore, this data suggests that the Z-DNA binding domain of ZBP1 is required for its function in promoting metastasis. We included this new data in our revised manuscript (Fig. 5e).

FVB/NJ mice were implanted with the MVT-1 ZBP1 KO with reconstituted WT ZBP1 or $Z\alpha 2$ mutant. Left panel shows the representative images of H&E stained lung sections from mice showing lung metastasis. Scale bar, 2 mm. Right panel shows the quantification of metastatic foci in lungs from mice at 5-week post transplantation.

Regarding whether ZBP1 is required for metastasis in other “metastasis” models, we did not use the B16 model as in this model, B16 cells normally are *i.v.* injected into mice. As shown in Supplementary Fig. 3n, when MVT-1 WT and KO cells are *i.v.* injected into mice, the lung metastasis of mice is similar. In addition, we found that blocking necroptosis in tumor cells

dramatically reduced the number of circulating tumor cells (CTCs) in blood (see below data with WT and MLKL KO MVT-1 tumors). Therefore, we think that ZBP1-mediated necroptosis plays a role in the intravasation process during metastasis. As we believe that elucidating the mechanism of ZBP-1-mediated necroptosis in promoting metastasis should be addressed in a follow-up study, we did not include the CTC data in our revised manuscript. To confirm the role of ZBP1 in metastasis in another model, we are using MMTV-PyMT mouse model. Our preliminary data with the straight ZBP1 KO MMTV-PyMT mice indicated that ZBP1 deletion dramatically reduced the lung metastasis in this model as well. As we are currently generating MMTV-ZBP1 conditional KO mice to confirm that ZBP1 deletion in tumor cells, not in other types of cells, is responsible for the reduced metastasis in this GEMM tumor model, we did not include this preliminary data in MMTV-PyMT model in the revised manuscript.

Clonogenic survival potential was compared between MVT-1 CRISPR CT and MVT-1 MLKL KO CTCs (circulating tumor cells) from mice blood serum post-implantation with MVT-1 CRISPR CT or MVT-1 MLKL KO at 4-week (left panel) or 5-week (right panel). Colonies were stained with crystal violet (0.02%). The graphs represent the relative percentage of colony numbers of MVT-1 CRISPR CT or MVT-1 MLKL KO CTCs.

Regarding the question “*Does the inhibition of necroptosis reduce inflammatory cytokines/chemokines that drive seeding into the lung?*”, as shown below, we found that blocking necroptosis in tumor cells reduces inflammatory responses in macrophages isolated from MVT-1 tumors. However, the overall levels of these cytokines in serum from mice with WT or MLKL KO tumors are very low and similar. While it is possible that the tumor necroptosis-regulated cytokine/chemokines may contribute to metastasis, considering the major difference of CTC levels in mice with WT and MLKL KO tumors, we are currently focused on investigating how necroptosis affects intravasation of tumor cells. But we discussed the possibility about the effect of necroptosis-induced inflammation in our revised manuscript.

Inflammatory cytokines are decreased in MLKL KO tumors
 Quantitative real-time PCR analysis of the relative expression of IL-1 β or TNF- α or IL-6 mRNA from macrophages isolated from the mice implanted with MVT-1 CRISPR CT or MLKL KO cells at 5 weeks.

IL-1 β or TNF- α or IL-6 levels were detected by ELISA in serum samples from MVT-1 CRISPR CT or MLKL KO implanted mice at 5-weeks.

Major comments:

1. The authors show that ZBP1 deletion reduces tumor necrosis but results in increased apoptosis. Is this IFN mediated or RIPK1-dependent apoptosis?

As shown in the new data below, extended IFN treatment can induce apoptosis in MVT-1 cells, but RIPK1 is not required for INF-induced apoptosis in MVT-1 cells. We included this data in our revised manuscript (Supplementary Fig. 4l). In addition, as we showed in Supplementary Fig 4d-g, extended hypoxia and nutrient deprivation also triggers apoptosis in these cells, we think that the increased apoptosis in ZBP1 KO cells may be caused by the combination of all of these factors: IFN, hypoxia and nutrient deprivation. We add this discussion in our revised manuscript.

Western blotting analysis of MVT-1 CRISPR CT or MVT-1 RIPK1 KO cells treated with IFN- γ for 36 or 72 hr using the indicated antibodies.

2. In several figures GD (or ZBP1 deletion) results in MLKL phosphorylation and RIPK3 degradation (Figs 4a,b;5b;Ext data Fig4a,b). This is unexpected as RIPK3 is required for MLKL phosphorylation.

We thank Dr. Kelliher for this keen observation and raising the issue. As shown in Supplementary Fig. 4a, GD triggers a gradual decrease of RIPK3 protein while MLKL phosphorylation is increasing. We confirmed that the decrease of RIPK3 protein is due to protein degradation as the presence of the proteasome inhibitor MG132 blocks the GD-induced decrease of RIPK3 (see new data below). When the necroptosis pathway is engaged, only certain percentage of RIPK3 protein is recruited to the necrosome complex as demonstrated by our and others' previous studies. Particularly, the phosphorylated MLKL will disassociate with RIPK3 and allow RIPK3 to phosphorylate additional MLKL. Therefore, while RIPK3 protein level decreases following GD, the remaining RIPK3 is sufficient to increase the accumulation of phosphorylated MLKL. We added this interpretation in our revised manuscript (Supplementary Fig. 4m).

Western blotting analysis of MVT-1 cells treated with GD with or without MG132 (2 μ M) for 24 hr using the indicated antibodies.

3. The authors treat with 5-azacytidine to increase RIPK3 and ZBP1 levels in the mammary tumor cell lines but this treatment has broad effects. Can IFN γ be used instead to at least increase ZBP1 levels? This may be sufficient to activate the pathway upon GD.

IFNs are potent inducers of ZBP1 expression as ZBP1 is an IFN inducible gene. However, as RIPK3 expression is inhibited by DNA methylation in MVT-1 cells, IFNs do not elevate RIPK3 expression (see Supplementary Fig. 6a). Therefore, IFNs are not sufficient to activate the pathway upon GD treatment. We discuss this point in the revised manuscript (discussion).

4. In Fig 4b, ZBP1 ko partially rescues tumor cell death upon GD and they show that STING does not contribute. Can the cell death be blocked with anti-TNF abs?

We tested the possibility that TNF signaling is involved in GD-induced apoptosis in ZBP1 KO cells. As shown in the new data below, neutralizing anti-TNF antibody has no effect on GD-induced cell death in ZBP1 KO MVT-1 cells. As GD leads to mitochondria damage, the remaining cell death is most likely due to the mitochondria-mediated intrinsic apoptotic pathway, which is in a slower kinetics comparing to necroptosis in WT cells. We added this discussion in our revised manuscript and the data as Supplementary Fig. 6b.

MVT-1 CRISPR CT or MVT-1 ZBP1 KO cells treated with 5-AD for 3 days, followed by treatment with GD for 30 hr with or without neutralized anti-TNF α antibody. Cell death analysis was determined by PI staining using flow cytometry.

5. The data that mito DNA binds ZBP1 is suggestive but not convincing based on assays used and the ZBP1 overexpression. Since the ZBP1 abs are limiting, can CRISPR be used to knock in a tag to the endogenous locus and IP done perhaps in IFN γ treated MVT cells?

Following the suggestion, we tried to knock in a FLAG-tag to the endogenous ZBP1 with the Mendenhall and Myers tagging and knock-in system³. As the selection gene of this system is under the ZBP1 promoter after being knocked in, and ZBP1 promoter needs to be induced by IFNs, making it difficult to isolate positive clones with the knock-in FLAG-tag in MVT-1 cells. Meantime, we tested several new ZBP1 antibodies that became available in the endogenous ZBP1 IP experiments. We found that an anti-ZBP1 antibody from Santa Cruz is able to pull down endogenous ZBP1 protein efficiently. With this antibody as shown below, we found that mtDNA binds to endogenous ZBP1 protein following GD. We added this new data in our revised manuscript (Supplementary Fig. 5f).

MVT-1 cells were treated with GD and/or IFN- γ . Innate ZBP1 was pulled down by IP with anti-ZBP1 antibody (Santa Cruz, sc-271483) from the cytosolic fraction, followed by PCR for detecting mitochondrial CytB and Nd2 (left panel). Right panel shows the pull-down efficiency of IP as a control for left panel under treatment.

Minor comments:

1. The ms could benefit from additional editing and to avoid overinterpretation. The data should be reported based on the mammary or other tumor cell line examined rather than making broad statements such as “RIPK1 is not required for tumor cells to undergo necroptosis during tumor development.” line 93-4.

Following the suggestion, we modified our statement accordingly by indicating in MVT-1 mammary tumors.

2. line 100: *spontaneous is not the correct term as this are GEMM*

We made the correction in our revised manuscript.

3. line 123: *specify MVT mammary tumor cell line or model rather than tumor development*

We made the changes in our revised manuscript.

4. *Does the expression of VEGF in MVT cell line influence the hypoxia assays or lung colonization? Can the effects of blocking necroptosis on metastasis be demonstrated in the B16 model?*

As shown in the attached Fig., the VEGF levels are not affected by ZBP1 deletion in MVT-1 cells. As shown In Supplementary Fig. 3n, when WT and ZBP1 KO cells are injected intravenously, as in B16 model, these cells migrated to the lung similarly. Also, we made similar observation about the effect of ZBP1 deletion on metastasis in MMVT-PyMT model.

Western blotting analysis of MVT-1 tumor or cells from CRISPR CT or ZBP1 KO was determined by using the indicated antibodies.

5. line 234 and elsewhere: rather than following GD treatment it would be better clearer to state under conditions of GD.

We made the changes in our revised manuscript.

References:

- 1 Sorvina, A. *et al.* Mitochondrial imaging in live or fixed tissues using a luminescent iridium complex. *Sci Rep* **8**, 8191, doi:10.1038/s41598-018-24672-w (2018).
- 2 Chen, J. *et al.* An in Situ Atlas of Mitochondrial DNA in Mammalian Tissues Reveals High Content in Stem and Proliferative Compartments. *Am J Pathol* **190**, 1565-1579, doi:10.1016/j.ajpath.2020.03.018 (2020).
- 3 Savic, D. *et al.* CETCh-seq: CRISPR epitope tagging ChIP-seq of DNA-binding proteins. *Genome Res* **25**, 1581-1589, doi:10.1101/gr.193540.115 (2015).

REVIEWER COMMENTS

Reviewer #1 (Remarks to the Author):

The authors have performed several of the experiments that I have requested and, consequently the work is improved. I do, however, still see some problems that in my opinion must be addressed with additional experiments.

- 1) The authors show that loss of ZBP1 reduces necrosis in growing tumours systematically by serial sectioning in only one tumour type. To show the generality of this finding, data from three different tumours should be presented.
- 2) The authors show impact of loss of Noxa on release of mitochondrial DNA only by using RNA interference. This must also be demonstrated in at least two different tumour cell lines by using Crispr/Cas9 technology due to the known off-target effects of RNA interference.
- 3) Given that loss of Noxa is shown to reduce MLKL activation, loss of Noxa should also reduce necrosis in growing tumours. This should be examined using Crispr/Cas9 to delete Noxa in at least three different tumour cell lines to demonstrate generality of the finding.

Reviewer #2 (Remarks to the Author):

I am happy with the revision and recommend the acceptance by Nature Communication. Junying Yuan

Reviewer #3 (Remarks to the Author):

The revised manuscript is significantly improved and my concerns are addressed. In particular additional evidence of mt DNA release and genetic evidence that DNA binding region of ZBP1 may be required for MVT-1 "metastasis". Additionally contributions of IFN and TNF were addressed. I now find the revised ms acceptable for publication in Nat Comm

Responses to the reviewers

Reviewer 1

The authors have performed several of the experiments that I have requested and, consequently the work is improved. I do, however, still see some problems that in my opinion must be addressed with additional experiments.

- 1) *The authors show that loss of ZBP1 reduces necrosis in growing tumours systematically by serial sectioning in only one tumour type. To show the generality of this finding, data from three different tumours should be presented.*

Response: In our revised manuscript, following reviewer 3's suggestion, we have made it clear through our manuscript that this work is about ZBP1 involvement in tumor necroptosis in MVT-1 breast cancer model. We do not generalize our findings to all types of solid tumors in this work. It is not clear to us if the reviewer wants us to perform additional experiments in three different pairs of WT and ZBP KO MVT-1 tumors or three different types of solid tumors. In the first cases, we performed additional serial section study in two additional pairs of WT and ZBP1 KO MVT-1 tumors. As shown in Suppl. Fig. 3d of our revised manuscript, the results of all three different pairs of tumors showed that the necrotic areas are significantly smaller in ZBP1 Ko tumors compared to WT tumors. (As we normally cut tumors in the middle, one half for Western blotting and the other half for tissue section, we serial sectioned one pair of these half tumor samples. The second pair of whole tumors were collected at 40 days after transplantation due to their slower growing to the size of 1.5 cm in diameter, this pair of tumors have more necrosis in general). These new data further supported that the serial cutting is not necessary because statistically analyzing 4-6 similar sized WT and KO tumors sectioned in the same way (the longest diameters of the tumors) will accurately reflect the levels of tumor necrosis in WT and ZBP1 KO tumors. Serial sectioning study does not add any further value/evidence to support our findings.

In the latter case (in three different types of tumors), it will take at least 1-2 year to generate ZBP1 ko tumors in three different types of tumor models. As discussed above, we do not generalize our findings to all of solid tumors, this suggestion is not necessary and not feasible for our current work about the role of ZBP1 in tumor necroptosis in MVT-1 breast tumor model.

- 2) *The authors show impact of loss of Noxa on release of mitochondrial DNA only by using RNA interference. This must also be demonstrated in at least two different tumour cell lines by using Crispr/Cas9 technology due to the known off-target effects of RNA interference.*

Response: As we indicated in our earlier point-to-point responses to reviewer 2 and 3, our data on NOXA just indicated that NOXA may be involved in mediating mtDNA release under GD condition. Understanding the role of NOXA in tumor necrosis or necroptosis is not in the scope of this current work and should be addressed in our future study. When we investigated the possible involvement of NOXA in mtDNA release, we used both siRNA and shRNA to knock down NOXA. Among the four shRNAs used in our experiments, shRNA #1 is most effective in NOXA knockdown (Suppl. Fig 5i). The shRNA knockdown of NOXA clearly blocked GD-induced MLKL phosphorylation (Suppl. Fig 5i). As shRNA is slightly less effective in knocking down NOXA expression, comparing to siRNA knock down, we did not include the NOXA shRNA results in our original manuscript. Since this reviewer

questioned the possible off-target effect of siRNA, we include the NOXA shRNA data in our newly revised manuscript to show that both NOXA siRNA and shRNA effectively knocked down NOXA and blocked GD-induced mtDNA release and MLKL phosphorylation. Also following the reviewer's suggestion, we confirmed the role of NOXA in GD-induced mtDNA release and MLKL phosphorylation in three additional tumor cell lines, B16, mouse breast cancer cell Met-1 and human MCF7 cells. These new data are included in Suppl Fig 5j-m of our revised manuscript.

The reason that we did not use sgRNA to knock out of NOXA is that we tried three different anti-NOXA antibodies (ProSci, 2437; abcam, ab13654; Novus Biologicals, NB100-56368) and found that none of them work by using overexpressed NOXA as a control. Without a working antibody, it is very complicated and difficult to generate and confirm sgRNA KO in somatic cells. Therefore, we used siRNA and shRNA to knock down NOXA in our study.

3) *Given that loss of Noxa is shown to reduce MLKL activation, loss of Noxa should also reduce necrosis in growing tumours. This should be examined using Crispr/Cas9 to delete Noxa in at least three different tumour cell lines to demonstrate generality o the finding.*

Response: Because NOXA is known to be involved in multiple aspects of mitochondrial functions and biology, such as inhibiting MCL1 and BCL2 functions, and the levels of NOXA expression is critical for tumor growth (Cancer Letters, Volume 447, 10 April 2019, Pages 12-23; J Oral Pathol Med. 2019;48:52–59). As tumor growth and mitochondrial functions are key factors in regulating the onset and the kinetics of tumor necrosis in addition to mtDNA release, the role of NOXA in tumor necrosis and necroptosis will be a complex and could not be evaluated by simply examining necrosis as the reviewer suggested. The role of NOXA in tumor necrosis and necroptosis will be systematically studied in our future work. Practically, as discussed above, without a good anti-NOXA antibody, we could not generate NOXA KO tumors with CRISPR/Cas9 technology.

REVIEWERS' COMMENTS

Reviewer #1 (Remarks to the Author):

The authors have not done the additional experiments that I have requested, such as those needed to prove that Noxa is critical for tumor cell necrosis. In the absence of such new experiments, the conclusions that the authors have reached are not supported sufficiently.

Reviewer #1 (Remarks to the Author):

The authors have not done the additional experiments that I have requested, such as those needed to prove that Noxa is critical for tumor cell necrosis. In the absence of such new experiments, the conclusions that the authors have reached are not supported sufficiently.

Response: as discussed in our previous response to the reviewer's comments, because NOXA is known to be involved in multiple aspects of mitochondrial functions, the role of NOXA in tumor necrosis and necroptosis will be a complex and could not be evaluated by simply examining necrosis as the reviewer suggested. The role of NOXA in tumor necrosis and necroptosis will be systematically studied in our future work. To make this point clear, we included the extended discussion of this issue in our revised manuscript.

In addition, we revised our manuscript by following the editor's extended comments and editorial requirements as highlighted in our revised manuscript.